# Modeling and Analysis of Drill String–Casing Collision under the Influence of Inviscid Fluid Forces

**Bernard Xavier Tchomeni Kouejou** *, **Desejo Filipeson Sozinando** and **Alfayo Anyika Alugongo**

Department of Industrial Engineering, Operation Management and Mechanical Engineering, Vaal University of Technology, Vanderbijlpark 1911, South Africa
* Correspondence: bernardt@vut.ac.za

**Abstract:** The current study aims to study the drill string–casing system operating in an inviscid fluid under imbalanced and rubbing damage types. The Navier–Stokes equations were linearized to establish the hydrodynamic forces surrounding the drill string and resulted in a five-dimensional system of nonlinear differential equations. To ensure the accurate acquisition of friction characteristics in a fluid medium, a nonlinear wavelet synchronized transform (NWSST) technique was enhanced based on the denoised wavelet hard thresholding algorithm to extract the features of the rubbing system. The developed model was verified through various test conditions, and the extracted data tests show that the frictional impact proves sufficient to modify the dynamic behavior of the drill string throughout the energy concentration with a slight shift above and below the resonant frequency. It was shown by simulation that the vibration of the submerged drill string system potentially enhanced highly undesirable hidden vibrational frequencies that led to a disturbed and chaotic 3D orbit pattern vibrational response. The experimental results show how vibration analysis combined with the synchrosqueezed technique can identify the condition of the drill string system even under harsh operating conditions and demonstrate that fluid enables the drill string system to rotate with minimum friction.

**Keywords:** axial, lateral, and torsional vibration; fluid–drill string interaction; 3D rubbing-caused impact; rotary drill string systems; wavelet synchrosqueezed

## 1. Introduction

Drill string machines are typically used to drill up to several hundred meters into the ground as well as into the sea. The vibrations of the drill string due to recurrent contact with an external element disturb the optimal control of the pressure at the drill head and considerably compromise the quality of the drilling, in particular in the upper tens of meters. The drill string system, typically used in a non-Newtonian fluid (mud) where the viscosity can change with applied shear, is also used in a Newtonian fluid such as inviscid water (sea). Different fluid models, namely, the Navier–Stokes equations, have been used to describe the interaction of Newtonian fluids with various structures. The industry is constantly researching new techniques to predict and estimate the remaining useful life of machinery operations based on newly developed monitoring technologies. Most of the techniques used to extract distinguishing features from rotating machinery models consist of classifying different types of faults based on the available system information. An important study conducted by Ibrahim (2010) assessed the relationship between head loss and the rotation speed of the well according to the geometry of the well and the flow regime [1].

According to research in the literature, premature wear of rotors such as drill strings is a predominant cause of mechanical failure as well as the geological environment of marine structures, which can significantly alter the mechanical properties of the fluid interacting with the rotor mass of the rotor, making the fluid density near the working area extremely

uneven. Drill strings are designed for tight, hard-to-reach access. In addition, strong vibrations and axially induced bit excitations cause early deterioration of the drill string. The structural durability and degradation of a drilling system operating in a fluid medium are influenced by the characteristics of the fluid. The unwanted drill string vibrations waste some of the energy that should be sent to the drill bit [2]. Monitoring the condition of drill rods is difficult due to their size and accessibility; therefore, failure prediction is one of the most complicated and misunderstood difficulties a drilling operator can face. Although the drill string system normally operates in a non-Newtonian fluid medium, there is little research on the behavior of the drilling system operating in a Newtonian fluid [3]. Modeling and monitoring a drill string interacting with an inviscid fluid are two of the most challenging topics in mechanical science that have piqued the interest of professionals in the field. Mud damping, drill–wellbore contact, stochastic bit–rock interaction forces, and various external sources all contribute to the complexity of the dynamic responses of the drill string [4]. Unfortunately, due to their use in a variety of locations and accessibility issues, drilling parameters (such as feed force, drill bit diameter, rotational speed, etc.) significantly affect the torque of the drill, and the underlying mechanisms have not yet been studied in depth. However, further research is needed to determine how torque responds to changes in these mechanical characteristics. Many field applications relying on vertical rotors have been deployed in a variety of industries, including oil and gas exploration and petroleum drilling engineering. Any exciting force can cause a vertical drilling system to vibrate and react. The drill rods are always in a state of longitudinal compression and inclination at parametric resonance under axial excitation. As a result, the impact of these drill strings on the borehole wall is substantially increased. Impact force fluctuates rapidly during a collision. Thus, a lateral stress wave will arise and propagate in the drill string. In practice, there is a high risk of friction between the rotor stabilizer and the static section of the machine for machines such as a rotating drill string [5]. Friction causes mechanical deformation of the rotor at the point of contact in this state. More intense contact, on the other hand, causes the drill string to heat up and lose stiffness over time, which can lead to catastrophic machine failure. As a result, condition monitoring of such a system is required, which is often focused on feature extraction for fault identification in the time or frequency domain. Gulyayev et al., studied the quasistatic stability of an elongated drill string while subjected to torque, axial load, rotational inertial force, and internal mud flow [6]. The inherent frequencies and mode forms of the elongated drill string were presented in their model. Additionally, these features are often load and speed sensitive, and they cannot reliably provide any information about the component problem that has occurred. To address vibration problems in vertical rotors, numerical simulation is essential as it pinpoints the root cause of problems in complex machinery and helps identify parts that need special attention. Therefore, minimizing downtime and maximizing equipment life can be monitored. However, since the majority of signals in mechanical engineering are transient and nonstationary, it also helps in discovering suitable solutions for a variety of nonlinear parameter applications by influencing structural resonance and achieving the best possible results.

Methods such as the finite element analysis model of a rotating drill string [7], the Euler finite difference technique [8], the method of disturbance [9], Fehlberg's fourth-to-fifth-order adaptive Runge–Kutta method [10,11], and the synchrosqueezing technique [12,13] were applied to reduce and solve the resulting algebraic equations of the drill pipe model. However, a comprehensive analysis of these issues remains a challenge due to their multidisciplinary nature. Despite the challenges of drill string model formulation, modeling is still considered a powerful method to study vibration propagation along the drill string and to analyze suspected vibration-related failures by decreasing the unwanted string vibration of stems. However, the drill string is not simply a bundle limited to vibrating in the air with mere external forces, but it is capable of operating in a fluid medium with recurrent drill string–casing contact. Many other factors such as fluid pressure, lateral torsional vibration of the drill string, axial vibration of the drill string, the eccentricity of the drill rods, and friction due to the flexibility of the drill string

affect the process of drilling. The extraction of drill string characteristics under such operating conditions is usually not measurable in real time, which makes their study one of the difficult and emerging problems in engineering fields. Several works have been conducted on drill string vibrations, and various contributions and simulation techniques have been made regarding the efficient operation of drilling under bit–rock interactions. Since the casing–drill string interaction term is highly nonlinear and unpredictable, mathematical models have been proposed in the literature for the rock–bit interaction, but they do not reflect the dynamics of the drill string under vibration because the lateral/torsional and axial vibrations are strongly coupled.

Thus, the objective of this study is to functionally evaluate the effect of inviscid fluid on the friction of a rotating drill string system during casing and bit–rock interactions. To assess these effects, the model of [14] is adopted to compare the dynamic response with and without the fluid–structure interaction mode. The drill string model is also improved based on [15] to include the effects of higher borehole wall friction and fluid propagation around the lateral/torsional/axial deflection of the train drilling. This work's contribution is considering a multi-degree-of-freedom forced excitation parametric drill string with dry and wet friction-induced vibrations in coupled mechanical oscillators. The current study results in original mathematical descriptions of the type of drill string systems and the corresponding vibration analysis through numerical simulation methods and experimental validation. The most important expected outcomes of this work include a fuller knowledge of dynamic phenomena occurring in force-excited drill string systems with an account of the energy source properties. This may require advanced and precise equipment to measure and diagnose vibration parameters using an orbit model, frequency spectrum, and wavelet synchrosqueezing techniques. The research conclusions of this work offer theoretical guidance for detecting subterranean structures and preventing failures in rotating dynamic systems. The results will also provide a reference for controlling drilling parameters in a fluid medium.

The remainder of this paper is organized as follows: in Section 2, theory on the modeling of the drill string system with a coupled axial/lateral/torsional displacement is established in detail, and the review and the principles of nonlinear frictional drill string–casing contact under the influence of full torque applied on the drilling system are expressed. In Section 3, the mathematical model of the inviscid fluid is established and coupled with the vertical rotor equation to generate the governing equations of the fluid–rotor system. In Section 4, the practical implementation is described, and the impulse extraction algorithm is presented. In Section 5, the performance of the method in terms of orbit model, frequency spectrum, denoise robustness, and energy concentration is verified by the drill string fault simulation signal. Real signal analysis is also extracted based on the proposed method applied to complex drill string fault cases, such as variable speed conditions and multipoint defects. Finally, the conclusion is obtained in Section 6.

## 2. Mathematical Modeling of the Drill String System

In reality, drill strings vibrate more often as a combination of all basic modes (lateral, torsional, and axial deflection), and the coupling of all these vibration modes could make the problem quite complex to study. In this formulation, it is assumed that (1) the drill string is flexible, isotropic, and homogeneous; (2) the upper transmission rotates with an initial angular speed $\Omega$ and provides a torque $T_M$ (neglecting the damping and friction torques at the upper transmission); and (3) the deflection of the drill string is produced by the displacement of eccentric centrifugal forces from the centerline. It is further assumed that the internal damping and flow-induced forces are taken into account at this stage. The Lagrangian method is used to model the drill string used in rotary operations; therefore, to perform the analysis model, it is also assumed that only the stabilizer contacts the borehole wall at a specific point as shown in Figure 1a. To simplify the drill string structure, Figure 1b shows a simplified axial/lateral/torsional model of a conventional vertical drill string model operating in an inviscid fluid. Understanding the drill string vibrations and their correct interpretation for different types of formations in the real-time drilling process

would help optimize the drilling vibration through automatic monitoring and control parameters. For a better understanding of the geometrical structure of the mechanical model, the corresponding simplified plane model is constructed in Figure 1b. The simplified equivalent drill string model is composed of a cylindrical shaft of the uniform section with an eccentric disc at the midspan of the submerged well inside a fluid-filled container. The conventional vertical oil well drill string consists of the bottom-hole assembly (BHA), which consists of a drill collar, a heavyweight drill shaft, and a non-magnetic drill string stabilizer to prevent the drill string from under-balancing.

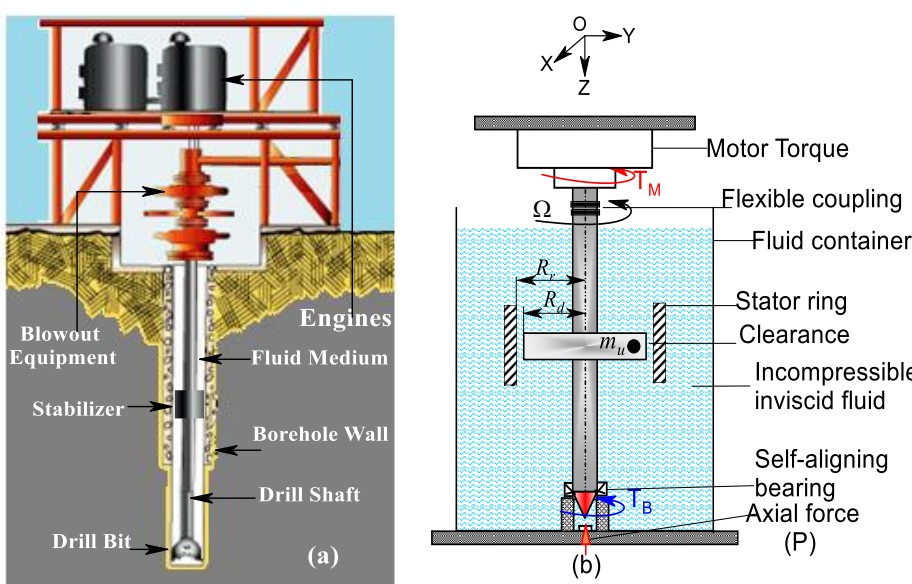

**Figure 1.** Typical rotary drilling rig. (**a**) Drill string model. (**b**) Simplified schematic model of the drill string.

In this paper, the BHA will also be referred to as a drill bit; the length of the BHA remains constant as the drilling operation progresses. It is also assumed that a section of the drill string is put under compression to apply the weight on the bit (WOB). The drill string is subject to compression due to the WOB, torsion due to rotation, lateral deflection due to the stabilizer eccentricity, and the bit-cutting effect. The motor torque is assumed to be constant and positive. The bit is supposed to be a simply supported rotor substantially subjected to the axial vibrations represented by a harmonic force. Shear and gyroscopic effects can be ignored since the structure is surrounded by a fluid with a low density compared to the rotating structure's material (usually steel). The equations for the lateral, axial, and torsion motion of the drill string can be derived by applying Lagrangian formalism to the pulling drum.

### 2.1. Energy Principle and Expression

The drilling system in this study considers only the directions of axial strain, bending, and torsion in the generalized shaft coordinates ($X$, $Y$, $Z$, $\theta$, and $\psi$). These include two horizontal displacements of the drill at the disk location ($X$ and $Y$), one axial displacement ($Z$), one rigid-body rotation ($\theta$), and two torsional deflection angles ($\psi$). The kinetic energy of the drill string disc is expressed as the sum of the translational and rotational mass imbalance and the kinetic energy of the motor system. The full expression for the derivation step appearing in this article is extensively developed in [14] and is therefore omitted for brevity and can be written as follows:

$$
\begin{aligned}
T = &\tfrac{1}{2}(m_u + M)\left(\dot{X}^2 + \dot{Y}^2 + \dot{Z}^2\right) + \tfrac{1}{2}J_D\left(\dot{\theta} + \dot{\psi}\right)^2 + \tfrac{1}{2}J_M\dot{\theta}^2 + \tfrac{1}{2}m_u e^2\dot{\psi}^2 + m_u e^2\dot{\theta}\dot{\psi} \\
&+\tfrac{1}{2}m_u e^2\dot{\theta}^2\left(1+\psi^2\right) - m_u\dot{X}\dot{\theta}\left[(e_x - \psi e_y)\sin\theta + (e_x\psi + e_y)\cos\theta\right] + m_u\dot{Y}\dot{\theta}\left[(e_x - \psi e_y)\cos\theta\right. \\
&\left.- (e_x\psi + e_y)\sin\theta\right] - m_u\dot{X}\dot{\psi}(e_x\sin\theta + e_y\cos\theta) + m_u\dot{Y}\dot{\psi}(e_x\cos\theta - e_y\sin\theta)
\end{aligned}
\tag{1}
$$

where $J_M$ is the inertia of the top rotary system, $M$ is the drill string mass, $J_D$ is the shaft–disc mass moment of inertia, $m_u$ is the imbalance mass, and the pairs, $e_x$ and $e_y$, are the components of $e$ in $x$ and $y$ coordinates. The inertias are interconnected by linear springs ($K_{XX}$, $K_{YY}$, and $K_{ZZ}$) which consist of three translational deformations ($X$, $Y$, and $Z$). Referring to a large bending deflection, axial movement, and torsional deformation $\psi$, the potential energy is expressed as:

$$V = \frac{1}{2}K_{XX}X^2 + \frac{1}{2}K_{YY}Y^2 + \frac{1}{2}K_{ZZ}Z^2 + \frac{1}{2}K_{\psi\psi}\psi^2 \tag{2}$$

where $K_{\psi\psi}$ is the system torsional stiffness, and $K_{XX}$, $K_{YY}$, and $K_{ZZ}$ are the stiffness coefficients associated with the system degrees of freedom given by:

$$K_{XX} = K_{YY} = K_{ZZ} = K_0 - \frac{P(t)\pi^2}{2L} \tag{3}$$

where $K_0$ is the modal drill string stiffness value, and $P(t)$ is an axially compressive force which is assumed to fluctuate harmonically around a constant mean value of the weight of the drill string $P_O$ and is written as [5]:

$$-P(t) = P_o + P_f \sin\Omega_f t \tag{4}$$

where $P_f$ and $\Omega_f$ are, respectively, the amplitude and the frequency of the axial force. The frequency of this periodic component is related to the frequency of the drilling system by $\Omega_f = n\Omega$, where $n$ is an integer indicating the bit factor, and $\Omega$ represents the rotary speed of the drill string. The Rayleigh dissipation function defined in the case of a viscous damper and the torsional damping attached to the system can be expressed as:

$$R = \frac{1}{2}C_T\dot{\psi}^2 + \frac{1}{2}C_{XX}\dot{X}^2 + \frac{1}{2}C_{YY}\dot{Y}^2 + \frac{1}{2}C_{ZZ}\dot{Z}^2 \tag{5}$$

where $C_T$ is the system torsional damping, and $C_{XX}$, $C_{YY}$, and $C_{ZZ}$ are the respective degree of freedom damping coefficients. Furthermore, viscous damping torque is taken into account at the upper drive system ($T_{en}$) and the bit ($T_{bt}$). Dry friction torque ($T_{Rb}$) is considered at the drill bit. The extended Lagrange formulation can be therefore used to derive the equation of motion of the damped drill string suspension system.

### 2.2. Mathematical Modeling of the 3D Rubbing Contact

During the experiment, the drill string, influenced by a centrifugal force during bending, always touched the borehole wall during drilling, and the borehole wall exerted pressure and frictional force on the drill string in response. As a result, two points of contact (axial bit load, shaft disc to casing wall) could, in the long run, cause premature wear and cracks. The collision effects between the shaft disc and the fixed casing are simplified to be the axial and lateral impact forces exerted on the outer surface of the casing, as shown schematically in Figure 2a. Figure 2b illustrates how the model proposed here, in contrast to others in the reviewed literature, accounts for the shaft's longitudinal sliding in the vertical direction Z to estimate the nonlinear forces during contact. It is suggested that the expression of the rubbing forces between rolling parts and the casing can be used to create a realistic rubbing model that is more complex than the typical radial bilinear model [16]. Although various friction schemes have been proposed, such as the Stribeck friction model, one of the most extensively used models to explain the friction phenomenon is the Coulomb friction law.

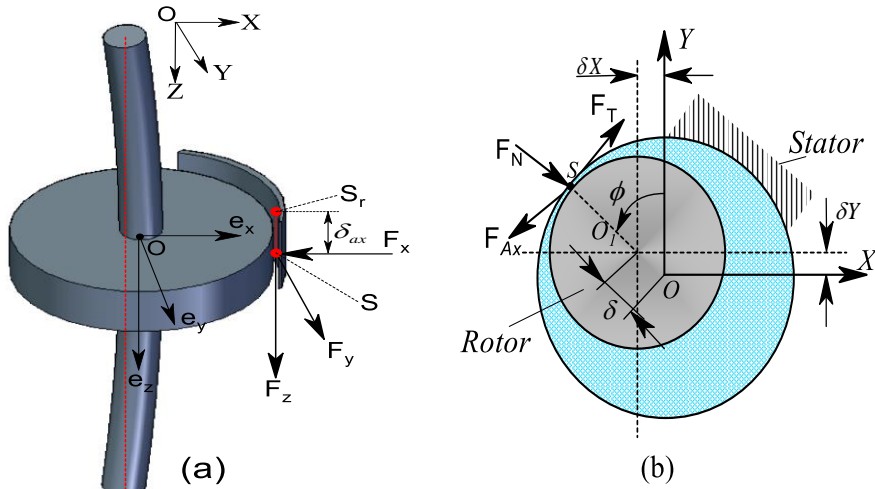

**Figure 2.** (**a**) Schematic of the rubbing system. (**b**) Model of the contact force.

In the cartesian coordinate system of X, Y, and Z (see Figure 2b), let S represent any known site of disc and casing contact. The axis Z is chosen to be orientated along the drill string's normal, while axes X and Y are oriented along the main axes of the cross-section. Let, $\vec{e}_X$, $\vec{e}_Y$, and $\vec{e}_Z$ be the unit vectors in the three-dimensional space X-, Y-, and Z-axes, respectively. The disc bends in the $\vec{e}_X$ and $\vec{e}_Y$ directions and also oscillates along the direction of $\vec{e}_Z$ (Figure 2a). The current position of the contact point S with respect to O as shown in Figure 3b is written by the vector. The contact then brings an added stiffness that is higher than the shaft stiffness and induces a restoring force. The expression of the radial impact force $F_N$ and the tangential rub force $F_T$ can be written by the Coulomb–Amontons law for friction as:

$$\begin{cases} F_N = (\Delta - \delta)K_s, & for \ \Delta \geq \delta \\ F_T = \mu F_N, & for \ \Delta < \delta \end{cases} \tag{6}$$

where $K_s$ is the stiffness of the casing in the radial direction, $\Delta = \sqrt{(S_X^r - S_X^s)^2 + (S_Y^r - S_Y^s)^2}$, which represents the radial relative displacement between the rotor (indexed by $r$) and the casing (indexed by $s$), $\mu$ is Coulomb's friction coefficient, and $\delta$ is the radial clearance between the rotor and the casing. $S_X$ and $S_Y$ are, respectively, used for horizontal displacements and $S_Z$ for vertical displacement. The mathematical phenomenon behind the nonlinearity due to the rubbing effect happens when the radial displacement becomes equal to the clearance between the drill string and casing ($\Delta = \delta$). The forces exerted in the plane between the drill string and the casing are therefore expressed as:

$$\begin{cases} F_X = -\cos\phi.F_N + \sin\phi.F_T \\ F_Y = -\sin\phi.F_N - \cos\phi.F_T \end{cases} \quad \text{with} \quad \begin{array}{l} \cos\phi = (S_X^r - S_X^s)/\Delta \\ \sin\phi = (S_Y^r - S_Y^s)/\Delta' \end{array} \tag{7}$$

This model is more generally used to model the rotor/casing contact in a Cartesian plane [16]. Substituting (6) into (7) gives the expression of frictional forces in the planes X and Y as:

$$F_{X\_rub} = -\frac{(\Delta - \delta)K_s}{\Delta}(X - \mu Y) \text{ and } F_{Y\_rub} = -\frac{(\Delta - \delta)K_s}{\Delta}(\mu X + Y) \tag{8}$$

The angle characterizing the position of the drill string relative to the casing is such that the relative displacement of the center of the drill string of the radius is $Z = \sqrt{S^2 - \Delta^2}$ which generates a vector of nonlinear forces $F = [F_{AX}, F_N, F_T]^T$ which are the static force loads in the radial (X and Y) and axial Z directions. In this case, with the contact force $F_N$, the inviscid fluid separating the rotating shaft and the fixed casing surfaces is assumed to induce relatively low friction. Therefore, the drill string and casing interact according to

Coulomb's unilateral contact law with dry friction, and the axial friction $F_{AX}$ is modeled conventionally by Coulomb's law, i.e., by $F_{AX} = \mu F_N$. A 3D rotor–casing contact friction problem is highlighted by the case study example in this section. The dissipative nonlinear force can be written in three dimensions, *X*, *Y*, and *Z*, in matrix form:

$$\begin{Bmatrix} F_{X\_rub} \\ F_{Y\_rub} \\ F_{Z\_rub} \end{Bmatrix} = -H(\Delta + \delta_{ax} - \delta)\frac{(\Delta + \delta_{ax} - \delta)K_s}{\Delta}\begin{bmatrix} 1 & -\mu & 0 \\ \mu & 1 & 0 \\ 0 & 0 & \mu \end{bmatrix}\begin{Bmatrix} X \\ Y \\ Z \end{Bmatrix} \tag{9}$$

where *H* is the Heaviside function applied along the lateral displacement (*X* and *Y*) defined as:

$$H(Z) = \begin{cases} 1, & \text{if } \Delta > \delta - \delta_{ax} \\ 0, & \text{if } \Delta \leq \delta - \delta_{ax} \end{cases} \tag{10}$$

The term $\delta_{ax}$ corresponds to the decrease in the radial clearance induced by a preload. For simplicity, the following set of assumptions can be made:

$$\begin{cases} \delta_{ax} < \delta & \text{if the load is not sufficient to ensure a rotor} - \text{stator contact,} \\ \delta_{ax} = \delta & \text{if the rotor just comes into contact flush with the stator,} \\ \delta_{ax} > \delta & \text{if the load is large enough to generate the prestress in the rotor.} \end{cases} \tag{11}$$

A significant point that appears for the first time in this study is the axial frictional force emanating from the axial force. It should be noted that the clearance is still well taken into account by this new formulation. Indeed, under the combined effect of this axial preload and a radial load, there may still be a loss in contact.

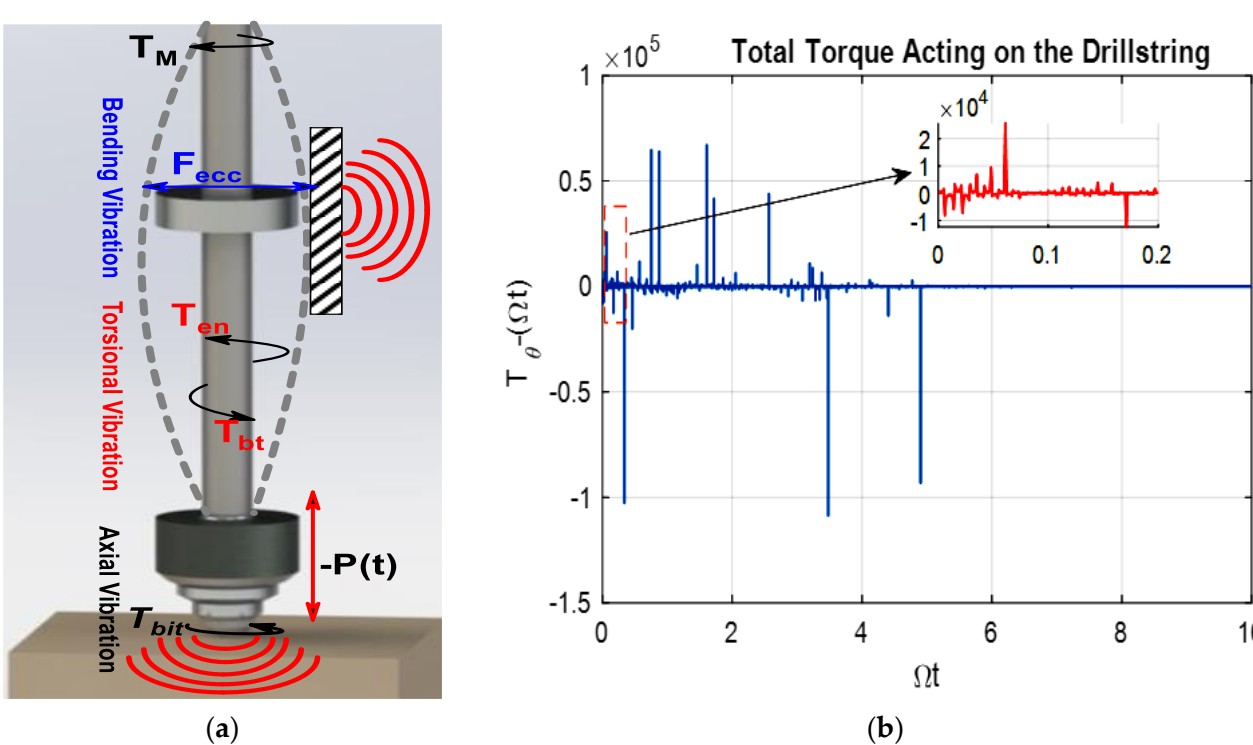

**Figure 3.** Schematic diagram of a bit–rock interaction during full operation and the full torque profile. (**a**) Full bit-rock torque contact. (**b**) Full bit-rock torque plot in drilling rig work.

### 2.3. Nonlinear Bit–Rock Interaction Model

The frictional force is recognized as the cause of the self-excited stick–slip phenomenon. To include the influence of torque in the drill string system, some mathematical torque equations are combined to simulate the dynamics of the drill string under various conditions. One of the major contributions associated with the existing models is the description

of some of the parameters involved, such as the presence of kinetic friction mechanisms, which leads to discontinuous differential equations and results in consistency with the field data. Therefore, the parameters that influence the friction mechanism, such as the angular displacement and angular velocity, must be considered in a proper stick–slip model (Figure 3). For this analysis, it is assumed that the bit is in continuous contact with the bottom and is laterally restrained. As a result, a viscous damping torque is considered both at the upper drive system ($T_{en}$) and the bottom hole of the drive bit ($T_{bt}$) caused by the interaction of the bit and rock level [17]:

$$T_{bit} = \left( \eta_{cb}(\omega_{Rb}) + (\eta_{sb} - \eta_{cb}) \times e^{-\lambda|\omega_{Rb}|} \right) \times (W_0 + W_1) R_{Rb} - \xi_b \times \omega_{Rb} \tag{12}$$

where $\lambda = 0.9$ is the decaying factor and $\omega_{Rb}$ is the angular velocity of the drill-bit (i.e., $\omega_{Rb} = \Omega$) $W_1 = k_f \chi_0 (1 - \sin(2\pi f_b t))$ stands for the amplitude value of the WOB and varies with the type of bit used, $\chi_0$ is the depth of the cut in one turn, and $f_b$ stands for the frequency, which is determined by the depth of the cut $\chi_0$ and the rate of penetration. In the above expression, $T_M$ operates alone when there is no bit contact with the ground. When frictional contact occurs, the system switches to the mixed total torque mode. The electrical motor's surface torque $T_M$ is represented by this symbol [18]:

$$T_M = -K_0 \theta + c_r \omega_{Rb} \tag{13}$$

The total nonlinear torque on the drill string bit ($T_{Tb}$) is modeled as the sum of the total torque. Unlike the engine torque, the friction contact is modeled as a dynamic lateral and axial discontinuous dry friction contact ($T_{Rb}$) expressed using Equation (9) by:

$$T_{Rb} = F_{Z\_rub} R_{Rb} = -H(\Delta + \delta_{ax} - \delta) \frac{(\Delta + \delta_{ax} - \delta) K_s}{\Delta} \mu Z \times R_{Rb} \tag{14}$$

The torque on the bit (TOB), generated in the ($\theta$, X, Y, Z) direction of the drill associated with the cutting torque $T_c$, can be expressed in the following equations:

$$T_{Tb} = \begin{cases} T_M & no\ contact \\ T_M + T_{bit} + T_{Rb} & if\ full\ contact \end{cases} \tag{15}$$

The data relating to an actual drill string design given in [17] are used in the simulations for the following: $Wo$ (base case) = 30 kN, $k_f = 23 \times 10^6$ N/m, $f_b = 0.005$ m, $R_{Rb} = 0.005$ m, $\eta_{sb} = 0.8$, and $\eta_{cb} = 0.45$.

The developed torque profile presented in Figure 3b shows that the fluctuations in the torque could have a detrimental effect on the drill bit and downhole equipment. The evolution of the torque during the stick–slip oscillation demonstrated that there are periodic fluctuations in the torque profile at startup around the mean value of 3400 Nm. The amplitude of this fluctuation is relatively low at the start but remains significant throughout the drilling process. Suddenly, the torque means the value drops to the minimum value, indicating that the drill bit is about to stick. At stick RPM, the bit stops momentarily causing the peak torque and TOB to build up almost linearly to a very high value, resulting in an extreme drop in torque, and the system becomes relatively stable.

## 3. Mathematical Modeling of the Inviscid Fluid System

In this section, the analysis is carried out, first theoretically, then experimentally, on a 560 mm shaft length immersed in a vertical rectangular rigid and impermeable container, which is estimated under the excitations of the drill string along the X-, Y-, and Z-axes. It is assumed that (1) under shaft excitation, the inviscid fluid motion is at a low Reynolds number and is entirely caused by the low amplitude vibrational motion of the drilling system, (2) the fluid is inviscid, incompressible, and during wet contact, the coefficient of friction and the wet coefficient are relatively the same, (3) the liquid can have a free surface when the inviscid fluid is considered, and (4) for sudden changes in section, the radius of

the disc is closer to the radius of the shaft ($R_{shaft} \approx R_{disc}$). The mass of the fluid contributing to the motion of the drill string is estimated using linearized Navier–Stokes equations that are simplified to Euler equations. As a result, the pressure distribution on the container surface is integrated to estimate the hydrodynamic forces acting on the drill string during forced excitation.

### 3.1. The Hydrodynamic Force under Lateral Excitation Force

The derivation of the hydrodynamic forces operating on a vertical container of depth h and width 2 W under sinusoidal lateral excitation is established under translational and pitching excitations. The drill string of length L is partially submerged in the reservoir filled with an inviscid fluid, as shown in Figure 4b. Rising to a dynamic imbalance, the drill string's center of mass is then deflected, which causes the harmonic excitation of the (x and y) axes by:

$$x(t) = X_0 \cos \Omega t \text{ and } y(t) = Y_0 \sin \Omega t \tag{16}$$

where $X_0$ and $Y_0$ are the excitation amplitudes, and $\Omega$ is a cyclical frequency in cycles per unit of time much lower than its fundamental frequency ($\omega \gg \Omega$) so that the liquid oscillates at exactly the excitation frequency.

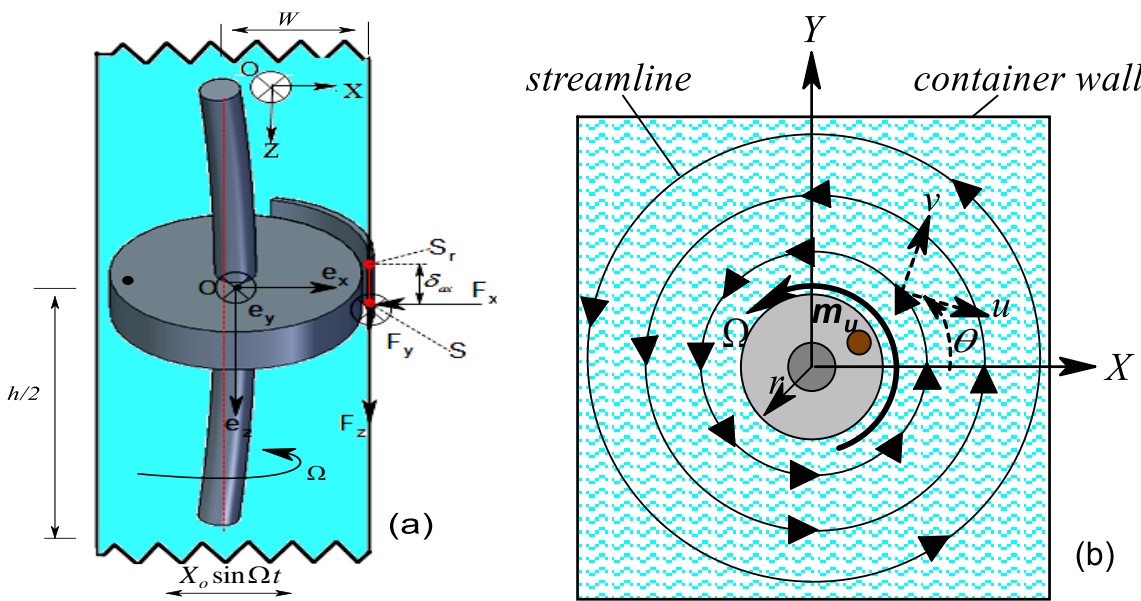

**Figure 4.** (**a**) Rectangular tank subject to harmonic excitation; (**b**) fluid profile around the shaft.

The following equations represent the linearized form of the fluid field equations for irrotational, inviscid, incompressible flow motion under the assumption that the submerged shaft oscillations are moderated [14]:

$$\nabla^2 \widetilde{\Phi} = 0, \text{ within the domain of fluids} \tag{17}$$

$$\nabla^2 = \frac{\partial^2}{\partial r^2} + \frac{1}{r}\frac{\partial}{\partial r} + \frac{1}{r^2}\frac{\partial^2}{\partial \theta^2} + \frac{\partial^2}{\partial z^2} \tag{18}$$

Where $\widetilde{\Phi} = \widetilde{\Phi}(r, \theta, z, t)$ ($\widetilde{\Phi} = \widetilde{\Phi}(r, \theta, z, t)$) is related to the flow disturbances and depends on Eulerian coordinates and time. The presence of the drill string and the boundary conditions at the container boundaries are:

$$\left.\frac{\partial \widetilde{\Phi}}{\partial r}\right|_{r=W} = 0, \quad \left.\frac{\partial \widetilde{\Phi}}{\partial z}\right|_{z=h} = 0, \quad \left.\frac{\partial \widetilde{\Phi}}{\partial \theta}\right|_{\theta=0, \pi/2} = 0, \tag{19}$$

The combined free surface condition obtained from the kinematic condition is:

$$g\eta - \frac{\partial \widetilde{\Phi}}{\partial t} + \ddot{x} r \cos\theta = 0, \text{ at } z = \eta(r, \theta, t) \tag{20}$$

The vertical velocity of a fluid particle resting on the free surface, $z = \eta(r, \theta, t)$, must be equated to the vertical velocity of the free surface itself to give the linearized dynamic free surface condition:

$$-\frac{\partial \widetilde{\Phi}}{\partial z} = \frac{\partial \eta}{\partial t} \text{ at } z = \eta(r, \theta, t) \tag{21}$$

where $\eta(r, \theta, t)$ is the fluid surface's height from the unaltered free surface. Once the kinematic free surface Condition (21) and the dynamic Circumstance (20) with respect to time have been differentiated, the outcomes are obtained:

$$\frac{\partial^2 \widetilde{\Phi}}{\partial t^2} + g\frac{\partial \widetilde{\Phi}}{\partial z} = \ddot{x} r \cos\theta \tag{22}$$

The function $\widetilde{\Phi}$ satisfies the following conditions on the fluid boundary: $\partial \widetilde{\Phi}/\partial \eta = 0$, at the container walls and the free surface. $\dot{\eta}(r, \theta, t)$ is the vertical velocity of the free surface. The typical solution of the continuity Equation (18) subject to boundary Condition (19) can be expressed as follows:

$$\widetilde{\Phi}(r, \theta, z, t) = \sum_{n=1}^{\infty} [C_{1n}(t)\cos\theta + D_{1n}(t)\sin\theta] J_1(k_{1n}r)\frac{\cosh(k_{1n}(z+h))}{\cosh(k_{1n}h)}] \tag{23}$$

where $C_{1n}$ and $D_{1n}$ are time-dependent functions determined from the free-surface initial Condition (20), $J_1(.)$ is the Bessel function of the first order, $k_{1n} = \xi_{1n}/W$ are roots of $\partial J_1(k_{1n}r)/\partial r|_{r=W} = 0$, and $r$ is a Fourier-Bessel series expansion given in the following form:

$$r = \sum_{n=1}^{\infty} F_n J_1(k_{1n}r) \tag{24}$$

where $F_n = 2W/(k_{1n}^2 W^2 - 1)J_1(k_{1n}W)$. Introducing Equations (23) and (24) with the free-surface Condition (20) yields:

$$\sum_{n=1}^{\infty} [\ddot{C}_{1n}(t) + \omega_{1n}^2 C_{1n}(t) - \frac{\ddot{x} F_n}{\cosh(k_{1n}h)}] J_1(k_{1n}r)\cos\theta + [\ddot{D}_{1n}(t) + \omega_{1n}^2 D_{1n}(t)] J_1(k_{1n}r)\sin\theta = 0 \tag{25}$$

where the natural frequency of the liquid-free surface in a rigid rectangular tank $\omega_{1n}$ can be obtained if the functions are expressed as harmonic functions. By substituting Equation (23) into the homogeneous Equation (22), the natural frequency expression is derived and expressed as:

$$\frac{(\omega_{1n}^2 \cosh((h+z)k_{1n}) - g\sinh((h+z)k_{1n}))\sin(\omega_{1n}t)(\cos\theta + \sin\theta)}{\cosh(hk_{1n})} = 0 \tag{26}$$

The corresponding natural frequencies are given by:

$$\omega_{1n}^2 = g\xi_{1n}\tanh(\xi_{1n}h/W)/W \tag{27}$$

where $\xi_{1n}$ are the roots of $\partial J_1(k_{1n}r)/\partial r|_{r=W} = 0$, and Expression (25) is satisfied if the functions $C_{1n}$ and $D_{1n}$ fulfill the following differential equations:

$$\ddot{C}_{1n}(t) + \omega_{1n}^2 C_{1n}(t) = \frac{\ddot{x} F_n}{\cosh(k_{1n}h)} \quad \text{and} \quad \ddot{D}_{1n}(t) + \omega_{1n}^2 D_{1n}(t) = 0 \tag{28}$$

The steady-state solutions of Equation (28) are:

$$C_{1n}(t) = -\frac{\Omega^3}{(\omega_{1n}^2 - \Omega^2)} \frac{X_0 F_n}{\cosh(\xi_{1n}h/W)} \cos\Omega t \quad \text{and} \quad D_{1n}(t) = 0 \quad (29)$$

By substituting Equation (29) into Expression (23), the velocity potential function is:

$$\widetilde{\Phi} = -X_0\Omega\cos\theta\cos\Omega t \sum_{n=1}^{\infty} [\frac{2W}{(\xi_{1n}^2 - 1)}\frac{\Omega^2}{(\omega_{1n}^2 - \Omega^2)}\frac{J_1(\xi_{1n}r/W)}{J_1(\xi_{1n})}\frac{\cosh(\xi_{1n}(z+h)/W)}{\cosh(\xi_{1n}h/W)}] \quad (30)$$

The disturbed fluid function $\widetilde{\Phi}$ and the reservoir potential function $\Phi = -X_0 r\cos\theta\cos\Omega t$ are coupled ($\Phi = \widetilde{\Phi} + \Phi_0$) to obtain the total potential function expressed as:

$$\widetilde{\Phi} = -X_0\Omega\cos\theta\cos\Omega t \times \left\{ r + \sum_{n=1}^{\infty} [\frac{2W}{(\xi_{1n}^2 - 1)}\frac{\Omega^2}{(\omega_{1n}^2 - \Omega^2)}\frac{J_1(\xi_{1n}r/W)}{J_1(\xi_{1n})}\frac{\cosh(\xi_{1n}(z+h)/W)}{\cosh(\xi_{1n}h/W)}] \right\} \quad (31)$$

Integrating the pressure distribution throughout the rectangular tank's bottom and walls yields the hydrodynamic force. The following equation can be used to determine the pressure distribution in any area of the reservoir at a depth h; since the fluid is inviscid, there is no shear force, and the gravitational effects ($\rho g z$) can be disregarded:

$$p = \rho\frac{\partial\widetilde{\Phi}}{\partial t} = \rho\Phi_0\Omega\left\{ r + \sum_{n=1}^{\infty} [\frac{2W}{(\xi_{1n}^2 - 1)}\frac{\Omega^2}{(\omega_{1n}^2 - \Omega^2)}\frac{J_1(\xi_{1n}r/W)}{J_1(\xi_{1n})}\frac{\cosh(\xi_{1n}(z+h)/W)}{\cosh(\xi_{1n}h/W)}] \right\} \quad (32)$$

The fluid pressure on the wall occurs on the disc at $\theta = 0$, and $\Omega t = \pi/2$ is given by:

$$\frac{p_w}{\rho g W} = \frac{\Omega^2 W}{g}\left\{ 1 + \sum_{n=1}^{\infty} [\frac{2}{(\xi_{1n}^2 - 1)}\frac{\Omega^2 W/\xi_{1n}g\tanh(\xi_{1n}h/W)}{(1 - \Omega^2 W/\xi_{1n}g\tanh(\xi_{1n}h/W))}\frac{\cosh(\xi_{1n}(z+h)/W)}{\cosh(\xi_{1n}h/W)}] \right\} \quad (33)$$

Similarly, the fluid pressure on the bottom of the thank at $z = -h$, $\theta = 0$, and $\Omega t = \pi/2$ is:

$$\frac{p_b}{\rho g W} = \frac{\Omega^2 W}{g}\left\{ \frac{r}{W} + \sum_{n=1}^{\infty} [\frac{2}{(\xi_{1n}^2 - 1)}\frac{\Omega^2 W/\xi_{1n}g\tanh(\xi_{1n}h/W)}{(1 - \Omega^2 W/\xi_{1n}g\tanh(\xi_{1n}h/W))\cosh(\xi_{1n}h/W)}\frac{J_1(\xi_{1n}r/W)}{J_1(\xi_{1n})}] \right\} \quad (34)$$

The integer *n* is the circumferential mode representing the deformation of the rotor. Taking into account the oscillation of the basic mode *m* = *n* = 1 such that the bottom of the reservoir Z = −h is produced by integrating the pressure on the appropriate region of the boundary and the net components of the hydrodynamic force acting on the wall *r* = *W*/2:

$$F_Z = \int_{\theta=0}^{2\pi}\int_{-h}^{0} p\sin\theta W d\theta dz = 0 \quad \text{and} \quad F_b = \int_{\theta=0}^{2\pi}\int_{r=0}^{W} prdrdz = 0 \quad (35)$$

Resolving along $\theta = 0$, the force exerted by the fluid along the fixed-coordinate *X*- and *Y*-axes is:

$$F_{XY} = \int_{0}^{2\pi}\int_{-h}^{0} p\cos\theta W d\theta dz \Rightarrow F_{XY} = m_f X_0\Omega^2\sin\Omega t\left\{ 1 + \frac{2W}{(\xi_{1n}^2 - 1)}\frac{\Omega^2}{(\omega_{1n}^2 - \Omega^2)}\frac{\tanh(\xi_{1n}h/W)}{\xi_{1n}h} \right\} \quad (36)$$

where $m_f = \rho\pi h W^2$ is the total mass of the fluid.

$$\frac{F_{XY}}{\rho g R^2 X_0} = \pi\frac{h\Omega^2}{g} \times \left\{ 1 + \frac{2W}{(\xi_{1n}^2 - 1)}\frac{\Omega^2}{(\omega_{1n}^2 - \Omega^2)}\frac{\tanh(\xi_{1n}h/W)}{\xi_{1n}h} \right\}\sin\Omega t \quad (37)$$

These results suggest that a net force is applied along the direction of excitation, according to Equation (36). The pressure distribution is symmetrical with respect to the reservoir in the perpendicular direction, canceling out the integration of the latter. There is no force in the direction of excitation produced by the pressure at the tank's bottom. It should be highlighted that the introduction of notions of resistance forces for such liquid-level systems enables the straightforward description of their dynamic properties. Therefore, when considering the vertical shaft's center coordinates along the *X*- and *Y*-axes:

$$X = W \cos \Omega t \text{ and } Y = W \sin \Omega t \tag{38}$$

For the first mode, the natural frequencies are obtained by setting $n = 1$; then, the hydrodynamic force can be rewritten as:

$$F_X = -M_{fl}\frac{d^2Y}{dt^2} \text{ where } M_{fl} = m_f X_0 \times \left\{ \frac{1}{W} + \frac{2}{(\xi_{11}^2 - 1)}\frac{\Omega^2}{(\omega_{11}^2 - \Omega^2)}\frac{\tanh(\xi_{11}h/W)}{\xi_{11}h} \right\} \tag{39}$$

Therefore, it is clear that in the scenario of a fluid surrounding a revolving drill string, the fluid's existence results in extra mass. The system's natural regime appears to have been altered, and the extra mass of the fluid is coupled to the shaft's inertial force as follows:

$$m_{XX} = m_{YY} = M + m_u + M_{fl} \tag{40}$$

To examine and comprehend, in particular, the influence of the fluid on the unsettling phenomena experienced during the friction and drilling operation, simulations and experiments were based on this straightforward formula.

### 3.2. The Equations of Motion of the Rotor–Casing Rub System Immersed in an Inviscid Fluid

To extract the features of the signal distributed in the time and frequency domains, an instantaneous energy spectrum distribution is performed. The five degrees of freedom (DoF) second-order differential equation regulating the mechanical system of a drilling rig in rubbing contact is established using the Euler–Lagrange formalism in an inviscid fluid and can be written as follows in matrix form:

$$
\begin{bmatrix} m_{\theta\theta} & m_{\theta\psi} & m_{\theta X} & m_{\theta Y} & m_{\theta Z} \\ m_{\psi\theta} & m_{\psi\psi} & m_{\psi X} & m_{\psi Y} & 0 \\ m_{X\theta} & m_{X\psi} & M + m_u + M_{fl} & 0 & m_{XZ} \\ m_{Y\theta} & m_{Y\psi} & 0 & M + m_u + M_{fl} & m_{YZ} \\ m_{Z\theta} & 0 & m_{ZX} & m_{ZY} & m_{ZZ} \end{bmatrix} \begin{Bmatrix} \ddot{\theta} \\ \ddot{\psi} \\ \ddot{X} \\ \ddot{Y} \\ \ddot{Z} \end{Bmatrix} + \begin{bmatrix} 0 & 0 & 0 & 0 & 0 \\ 0 & C_{\psi\psi} & 0 & 0 & 0 \\ 0 & 0 & C_{XX} & 0 & 0 \\ 0 & 0 & 0 & C_{YY} & 0 \\ 0 & 0 & 0 & 0 & C_{ZZ} \end{bmatrix} \begin{Bmatrix} \dot{\theta} \\ \dot{\psi} \\ \dot{X} \\ \dot{Y} \\ \dot{Z} \end{Bmatrix}
$$
$$
+ \begin{bmatrix} 0 & 0 & 0 & 0 & 0 \\ 0 & K_{\psi\psi} & 0 & 0 & 0 \\ 0 & 0 & K_0 - \Delta P & 0 & 0 \\ 0 & 0 & 0 & K_0 - \Delta P & 0 \\ 0 & 0 & 0 & 0 & K_0 - \Delta P \end{bmatrix} \begin{Bmatrix} \theta \\ \psi \\ X \\ Y \\ Z \end{Bmatrix} + \begin{Bmatrix} Q_\theta \\ 0 \\ Q_X \\ Q_Y \\ 0 \end{Bmatrix} = \begin{Bmatrix} T_{Tb} \\ T_M \\ F_X \\ F_Y \\ F_Z + F_g \end{Bmatrix} \tag{41}
$$

where $\Delta P = P\pi^2/2L - F_\psi\pi^3/2L^2$. The external excitation forces associated with each degree of freedom are expressed as $F_g$, the force vector induced by gravity, $F_\theta$, the input force from the motor, and $F_\psi$, the fluctuating axial force. $\{Q_n\}$ represents the nonlinear part, which is shown to be a function of the system rotational speed, the torsional deformation angle, and the mass imbalance. The drill string–fluid interaction provides the coupling between the fluid forces and the imbalanced lateral and torsional coupled rotor. It can be noted that due to the formulation of the stresses, the axial, torsional, and lateral vibrations are coupled. The coupling mass matrix between all degrees of freedom and the expression

containing the nonlinear terms and the possibility of periodic instability at very high shaft speeds is introduced in Equations (42)–(52) written in the form:

$$m_{\theta\psi} = m_{\psi\theta} = m_{\psi\psi} = J_D + m_u e^2 \tag{42}$$

$$m_{\theta\theta} = J_M + J_D + m_u e^2 \left(1 + \psi^2\right) \tag{43}$$

$$m_{\theta X} = m_{X\theta} = -m_u \left[ (e_x - \psi e_y) \sin\theta + (e_x \psi + e_y) \cos\theta \right] \tag{44}$$

$$m_{\theta Y} = m_{Y\theta} = m_u \left[ (e_x - \psi e_y) \cos\theta - (e_x \psi + e_y) \sin\theta \right] \tag{45}$$

$$m_{\psi X} = m_{X\psi} = -m_u \left[ e_x \sin\theta + e_y \cos\theta \right] \tag{46}$$

$$m_{\psi Y} = m_{Y\psi} = m_u \left[ e_x \cos\theta - e_y \sin\theta \right] \tag{47}$$

$$m_{XX} = m_{YY} = m_{ZZ} = M + m_u \tag{48}$$

$$K_{\psi\psi} = K_T - m_u e^2 \dot{\theta}^2 \tag{49}$$

$$Q_\theta = 2 m_u e^2 \psi \dot{\psi} \dot{\theta} \tag{50}$$

$$Q_X = -2 m_u \dot{\theta} \dot{\psi} (e_x \cos\theta - e_y \sin\theta) - m_u \dot{\theta}^2 \left[ (e_x - \psi e_y) \cos\theta - (e_x \psi + e_y) \sin\theta \right] \tag{51}$$

$$Q_Y = -2 m_u \dot{\theta} \dot{\psi} (e_x \sin\theta + e_y \cos\theta) - m_u \dot{\theta}^2 \left[ (e_x - \psi e_y) \sin\theta + (e_x \psi + e_y) \cos\theta \right] \tag{52}$$

The governing equations are fully coupled nonlinear differential equations with time-varying coefficients. The time-varying coefficients are due to fluid–shaft excitation and axial coupling. Axial and transverse motions are coupled due to nonlinear elastic deflections, while the transverse motions in the $X$ and $Y$ directions are coupled due to hydrodynamic damping, shock impact, and frictional forces.

## 4. Description of the Test Bench and Experimental Approach

To validate the theoretical model, an experimental setup is developed using a miniaturized Bently Nevada rotor-kit 4. The test rig consists of a flexible vertical shaft with an imbalanced disc attached to it and connected to a variable-speed motor. The system is installed inside an 84.5 × 25 × 25 cm$^3$ rectangular container filled with non-viscous water. Engine speed is measured with a laser tachometer and the response is captured with devices such as probes and stainless-steel pressure sensors inside the water. The test bench set up for the experiment is described in the following section. The proposed rig design is focused on studying the rotational motion of a small-scale drill string, where axial and lateral vibrations are not recorded.

### 4.1. Experimental Setup Description

The photograph presented in Figure 5 was used to analyze the imbalance and rub impact in a fluid medium. The experimental test rig consists of a motor, flexible coupling, a drill string shaft, one disc, two journal bearings and bearing blocks, six proximity probes, a rub screw, a plexiglass container filled with water, and a data acquisition interface unit that is controlled by Ascent software. An interchangeable flexible steel shaft that transmits the rotational movement of the motor to the BHA is used to reproduce the drill rod. The coupled motor and rotor's shaft are simply supported at the top end by a wooden base inserted in a V-frame design that was chosen to provide better control of the dynamic stiffness properties of the housing. A vertical shaft of 560 mm in length and 10 mm in diameter carries a disc with a diameter and thickness of 75 mm and 35 mm, respectively, mounted at the position of 280 mm and driven by an electric motor incorporated with a shaft through a flexible coupling. The motor is placed on the rig platform, typically with a speed ranging from [0–1600 rpm]. It is used to maintain the desired WOB, bit drill, and pull out the top drive drill and requires an amplifier system to maintain proper speed control.

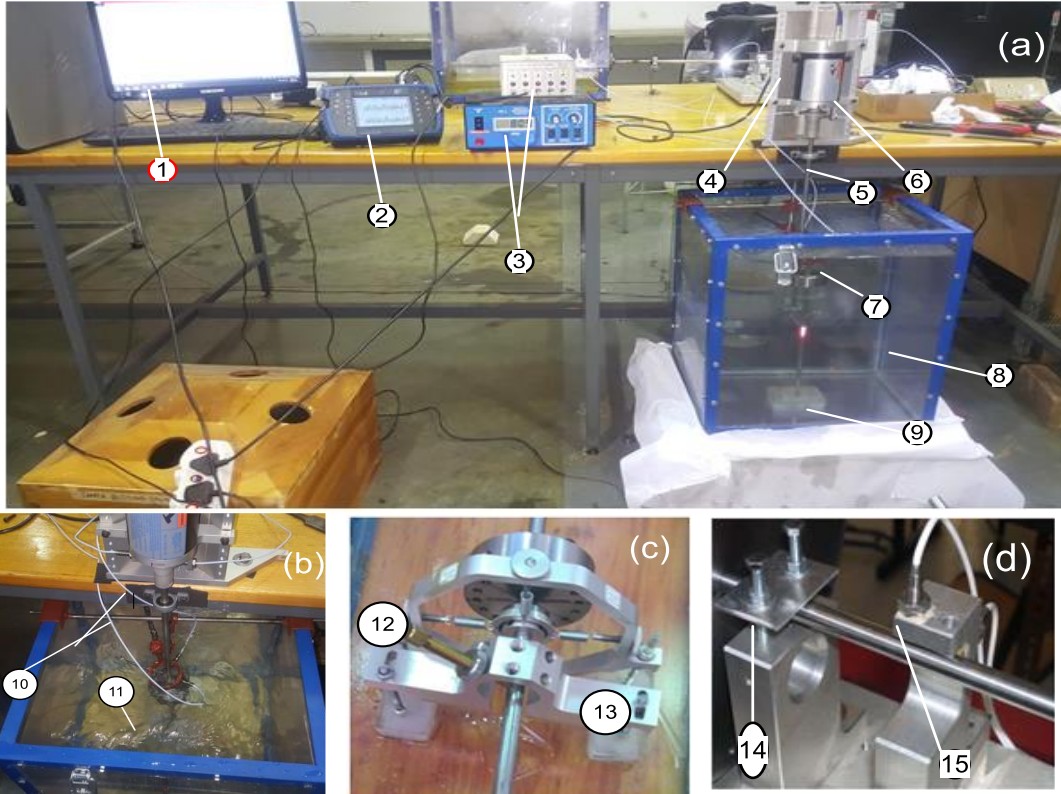

**Figure 5.** Miniature drilling rig model for experimental analysis. (**a**) 1-Computer screen, 2-ccout-2013, 3-speed control junction box, 4-frame support, 5-vertical shaft, 6-motor, 7-disc, 8-tank, 9-granitic rock sample; (**b**) 10-probe sensors, 11-water; (**c**) 12-rub screw, 13-rub housing; (**d**) 14, 15–90° mounted sensors.

The motor and the torque system are fixed on the base, and the stage is connected to the load table by the steel cable and the fixed pulley. When the weights are placed on the load table, different weights can be set to control the drilling feed force. The rub screw is used to rub the radial surface of the shaft at a distance of 240 mm from the bottom. The rub screw is adjustable to achieve various rub degrees and is secured in the mounting block with a locknut (corresponding to the reduction in the screw–shaft clearance area). Indeed, the lateral deflections of the shaft are created by a 3.2 g weight setscrew tightened in the disc serving as an imbalance mass to completely facilitate the recurrent screw–shaft contact. The experimental setup was adjusted to use real granitic rock samples in the drill tests, where a rotating handle under the experimental platform was designed to change the height of the sample to provide an additional WOB for the drill string. The vibration measurements are conducted by using a laser tachometer to assess the rotor's speed and six proximity probes and four stainless steel pressure sensors mounted in both orthogonal directions near the rub position and shaft support, as illustrated in Figure 5, to capture the vibration. The probes are connected to the data acquisition devices, which are in turn connected to the PC. The data acquisition interface unit operation is controlled by the Ascent-2013 software to collect and store vibration data. Then, the data are transferred from the Ascent software to Matlab for processing and analysis.

### 4.2. Fluid Model Properties

For a rectangular section, the damping factor is a function of the liquid's height and the tank's width. For the aforementioned expression, $\omega_{1n} = (g\,\xi_{1n}/W\tanh(\xi_{1n}h/W))^{1/2}$ of the liquid-free surface gets closer to being constant. A family of unit-step response curves

$F_X/\rho g X_0 W^2$ with the dimensionless variable $\xi_{1n}$ and the frequency $h\Omega^2/g$ is shown in Figure 6a,b.

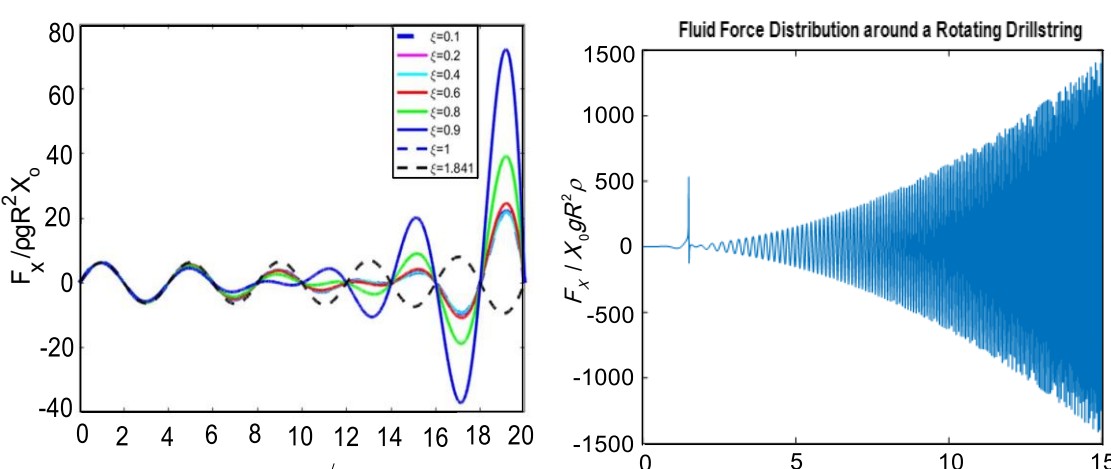

**Figure 6.** Dependence of the ratio of the hydrodynamic forces on the damping and frequency parameters. (**a**) Fluid forces for various values of $\xi$; (**b**) Fluid forces around the oscillating drill.

From Figure 6a, it can be seen that an underdamped fluid system between 0.1 and 0.9 gets close more rapidly than a critically damped or overdamped system. When increasing the damping parameters, the amplitude of the hydrodynamic force is exponentially increased away from the equilibrium position. The hydrodynamic forces acting on the rectangular container experience harmonic motion by overshooting the equilibrium position rapidly and exhibiting the fastest unstable response. Overdamped hydrodynamic forces are sluggish in responding to any damping increment, as observed by the dashed curves. However, Figure 6b presents the dependence of fluid forces on the lateral excitation frequency. From the foregoing analysis of the hydrodynamic forces, the fluid force is either critically damped or overdamped monotonically at the same magnitude. The downhole vibration of a drill bit, such as axial, lateral, and torsional vibrations, can cause premature tool failure. To optimize the drilling performance and extend the life of the drilling tools, controlling or mitigating downhole vibration is essential [19,20]. The natural frequencies of the five DoF kinematic chain are as follows: $f_{01}$ = 0 Hz, $f_{02}$ = 4.0756 Hz, $f_{0X}$ = 53 Hz, $f_{0Y}$ = 53 Hz, and $f_{0Z}$ = 187 Hz. The fluid drill string parameters used in this study are listed as indicated in Table 1.

**Table 1.** Data parameters used in rotary drilling simulation.

| Drill String Parameter | Value and Unit | Bearing Stiffness | Value and Unit |
|---|---|---|---|
| Length of the shaft (L) | 570 mm | Bearing shaft stiffness ($K_o$) | $7.35 \times 10^5$ Nm$^{-1}$ |
| Shaft diameter (D) | 10 mm | Casing stiffness ($K_s$) | $6.9 \times 10^7$ N/m |
| Disc radius ($Rd$) | 0.15 m | Lateral damping ratio ($\xi_l$) | 0.0287 |
| Disk moment of inertia ($J_D$) | 0.1861 kg.m$^2$ | Torsional damping ($\xi_T$) | 0.03 |
| Motor moment of inertia ($J_M$) | 10.36 kg.m$^2$ | **Disc** | **Value and Unit** |
| Shaft torsional damping ($C_T$) | 90 Nms/rad | | |
| Damping of the shaft ($C_0$) | 100.44 kg/s | Shaft–disc mass (M) | 16.845 kg |
| Friction coefficient ($\mu$) | 0.2 | Eccentricity mass (m$_u$) | 0.25 kg |
| Rotor–casing clearance ($\delta$) | 20 μm | Mass eccentricity (e) | 0.0011 m |
| Radial clearance ($\delta_{ax}$) | 60 μm | Disc thickness ($D_{disc}$) | 25 m |
| **Inviscid Fluid Properties** | **Units** | **Tank Parameter** | **Value and Unit** |
| Water ($T°$) | 20 °C | Tank width (W) | 0.3 m |
| Fluid frequencies ($\omega_{1n}$) | 6.01 rad/s | Excitation amplitude ($X_0$/W) | 0.166 |
| Fluid density ($\rho$) | 1004 kg/m$^3$ | Axial force magnitude ($P_f$) | 50 kN |
| Shaft–fluid angular speed ($\Omega$) | 4.8522 rad/s | Static weight component ($P_o$) | 100 N |

## 5. Numerical Simulation Results and Discussion

Based on Equations (16) and (41), the numerical simulation was conducted under the baseline and fault conditions presented as follows: 1. Baseline response: the imbalance drilling system is shown in both mediums. 2. Drill–casing rubbing condition: the periphery disc on the imbalanced shaft is supposed to have rubbing forces that are normal, tangential, and axial to it. 3. Submerged condition: in addition to imbalances and rubbing, the drill string system is simulated when immersed in an inviscid fluid in the lateral direction, the orbit of the shaft center, and the frequency spectrum of the signal. 4. For this paper, NWSST methods and IF have been developed to allow the signal processing of nonstationary signals and to simulate a vibration signal from an industrial model of multicomponent vibrations generated under a specific fluid regime.

### 5.1. Imbalanced Drill String System Response

The baseline data response of a simulation performed on an imbalanced drill string system during the passage through the first critical speed and the self-excited torsional vibration of the drill string is explored.

Figure 7b shows a drop in the torsional deflection signal, which keeps on gradually attenuating until the torsional vibration stops. Due to the nonlinear effects inherent in the drill string system problem, the oscillations do not grow infinitely; rather, during a collision, it is found that, in the case of deterministic friction without water, the component corresponding to the circumferential vibrations of the drill string disc is always dominant, and random friction creates impulsive forces which excite the transverse mode (Figure 7a). Figures 8 and 9 are an expanded view of the spectrum around the drill string speed up to two orders with the appearance of noise peaks in specific areas of the spectrum. The drill string hump of energy just around $66.3135X$ and $132.627$ orders in the air is possibly a structural half and critical resonance (Figure 8b). Contrary to the air medium, the structure in a fluid environment in Figure 9b shows an imbalanced fault characteristic frequency. The observation of the envelope spectrum in Figure 9b demonstrates a reduction in the vibration amplitude resulting in the soft fluctuation of the system. The vibration changes drastically in magnitude as a result of fluid pressure. Common changes in patterns (peaks, harmonics, noise, and orbits) can be observed. The presence of fluid reduces the number of peaks from the fundamental frequency and increases progressively after the amplitude of fluctuation from $0.84 \times 10^{-4}$ m to $1.11 \times 10^{-4}$ m in the inviscid fluid. The vibration responses displayed by the orbit patterns are distorted and cease to be pure harmonic signals, as shown in Figure 8a, contrary to Figure 9a, wherein the immersed fluid in the shaft center orbit in the $X$ and Y directions are multi-circular circles.

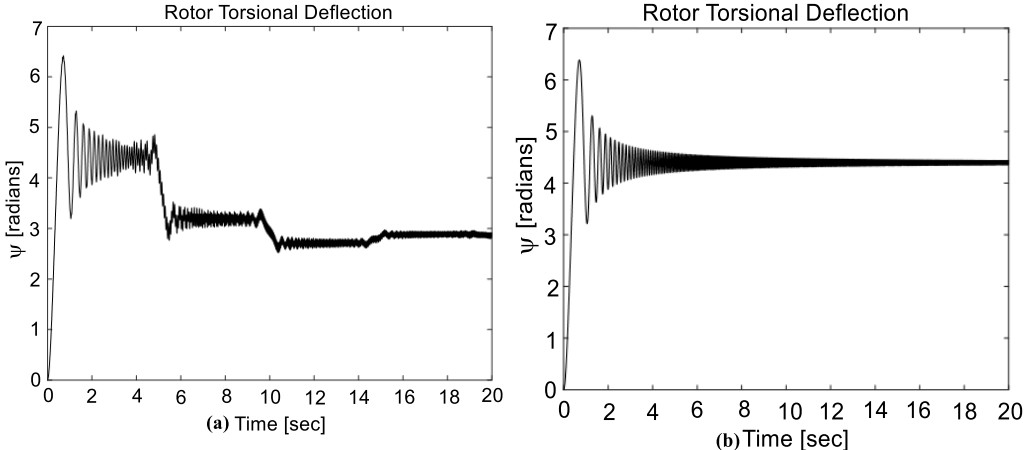

**Figure 7.** Torsional response of the imbalanced drill string-casing system with rub in an inviscid fluid. (**a**) Torsional in air; (**b**) Torsional in an inviscid fluid.

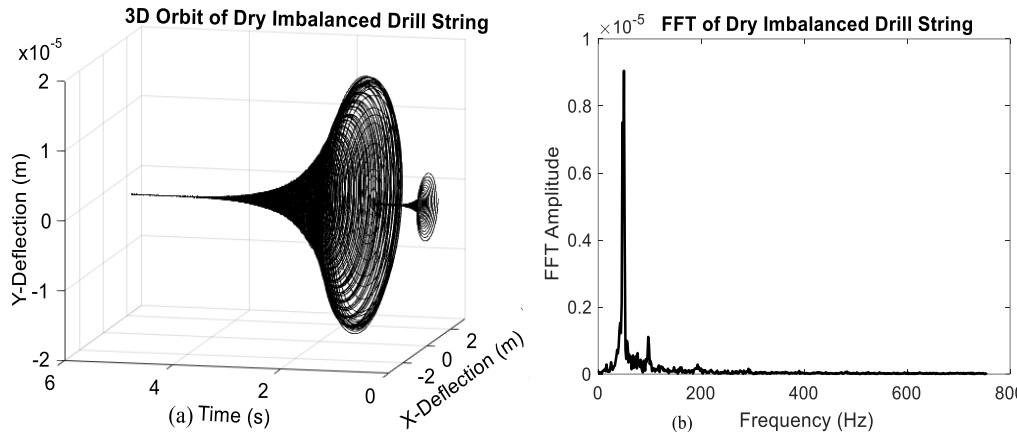

**Figure 8.** Response of the imbalanced drilling system while passing its first critical speed. (**a**) 3D imbalance response orbit in air; (**b**) FFT response of imbalanced drill in air.

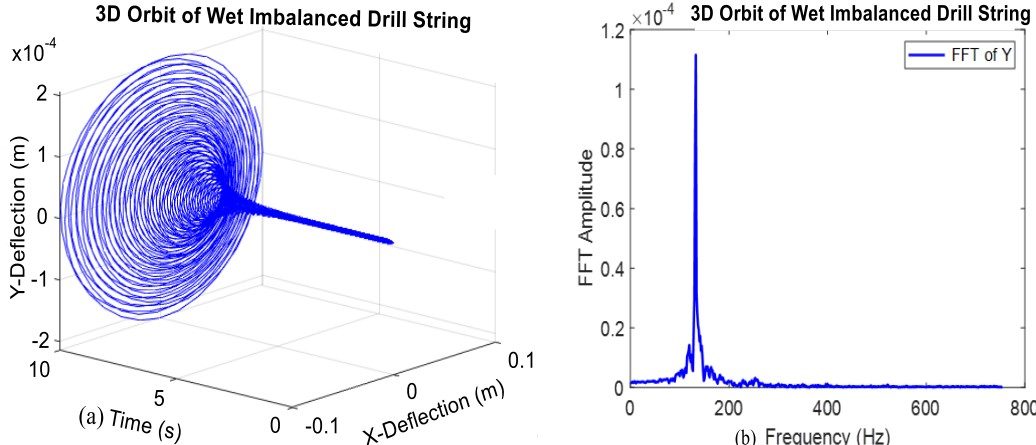

**Figure 9.** Response of the imbalanced drill string system immersed in an inviscid fluid at the first critical speed. (**a**) 3D imbalance response orbit in fluid; (**b**) FFT response of drill string in fluid.

**Observation 1:** The measured vibration level at 1$X$ depends on the stiffness of the machine mounting as well as the eccentric mass showing a greater 1$X$ than damping-mounted machines for the same degree of imbalance. Because of the fluid impact, the vibration level at the 1$X$ frequency decreases over a period of time. The more the fluid is introduced, the more likely that an imbalance will be detected. In any case, the direction in which the drill string is the least stiff is the direction of the highest 1$X$ level. The horizontal vibration will therefore typically be higher than the axial. However, what will the axial data reveal?

### 5.2. Response of the Imbalanced Drilling System and Diagnosis of the Rubbing Effects

Flexible coupling problems add multiple harmonics, such as imbalance and friction, producing a variety of symptoms in different environments; each case must be individually diagnosed. In the case of an imbalanced drill string with a rub, the effects of the fluid influence significantly the rub impact features. An interesting behavior observed in Figures 10 and 11 is the effect of the hydrodynamic forces on the system stability. As it is seen, the existence of rub on the imbalanced drill string causes instability in a highly disturbed drilling system, and this effect is intensified proportionally to the axial excitation force for a given rub clearance ($\delta = 6 \times 10^{-4}$ m selected). When the bit begins to interact, the system will inevitably be disturbed by the possibility that axial and lateral vibrations occur simultaneously, and the formation of their critical frequencies is noticeable. Eccentricity, in this case, produces a high vibration level at 1$X$ in the radial direction. Due to various nonlinear coupling terms, all critical frequencies are increased during contact conditions, and several resonance peaks are

observed as shown in Figure 10b. The effect of the permanent impact of the drill string on the casing generates multiple unwanted critical speeds, occurring at 75 rpm, 175 rpm, and 225 rpm (*1X, 2X . . . 5X*). These undesirable frequencies show that the rub on the vertical drill string with axial excitation forces would practically reinforce the high vibration of the system and cause its destruction. The results of the excitation due to the imbalanced drill string–casing contact and axial forces, in Figure 10a, generate a high unwanted fluctuation along with axial displacement (Z-axis), as shown in the 3D orbit plot where the rub impact is strongly irregular throughout the radial bilinear friction forces. From Figures 10a and 11a the 3D orbit vibration of the drill string experiences changes from being periodic to chaotic eventually. It is shown that, depending on the radial bilinear frictional forces and the axial friction, the drill string undergoes a transient phase marked by intermittent rebounds.

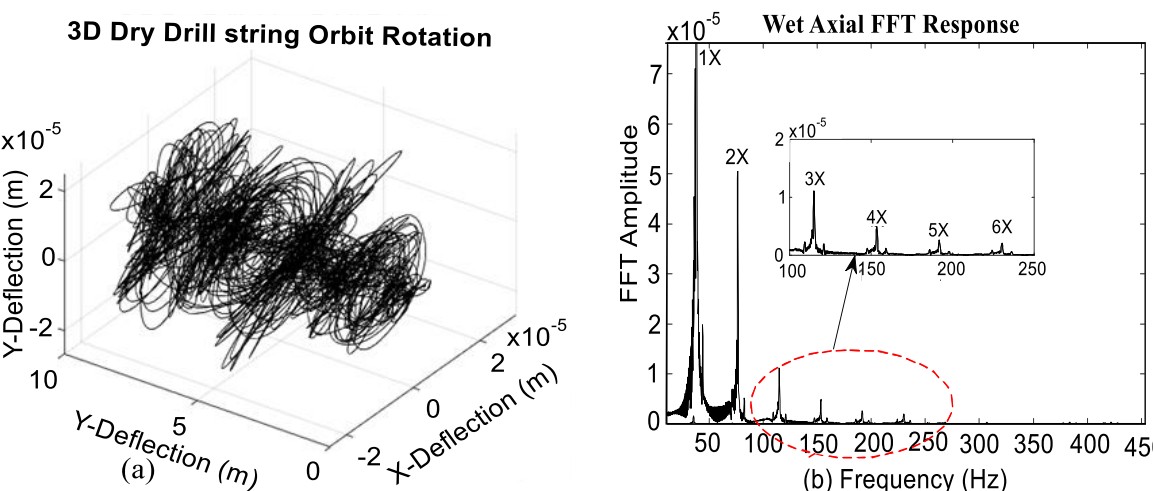

**Figure 10.** Response of the imbalanced drill string-casing system with rubbing. (**a**) 3D imbalanced drill string response orbit in air with rub; (**b**) FFT response of imbalanced and rubbed drill in air.

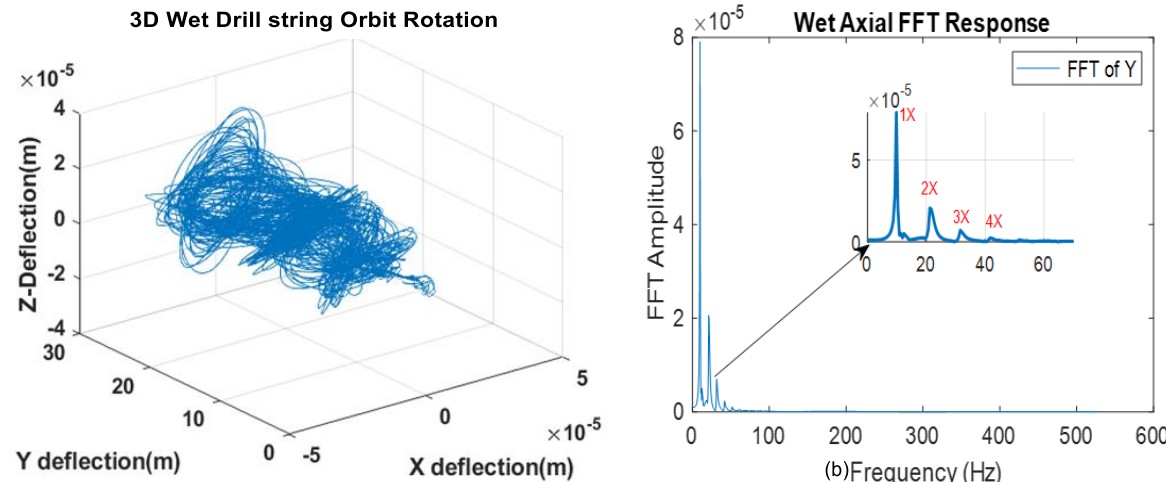

**Figure 11.** Response of the imbalanced drill string-casing system with rubbing in an inviscid fluid. (**a**) 3D imbalanced drill string response orbit in fluid with rub; (**b**) FFT response of imbalanced and rubbed drill in fluid.

It is apparent, comparing both Figures 10b and 11b, that when the hydrodynamic forces are applied, the vibration response decreases considerably with a reducing range of the first critical speed. The presence of dissipative effects such as hydrodynamic forces eventually settles into multiple limit cycle behaviors, as seen in Figure 11a. Figure 11a reveals that, depending on the radial bilinear frictional forces and the axial friction, the unstable contact is intermittent along the radial direction where the friction, despite the presence of fluid, remains

effective. This means that the hydrodynamic forces influence the motion of the drill string, as observed throughout the torsional deflection (Figure 7b), where the torsional motion becomes attenuated with a large amplitude in the presence of fluid.

**Observation 2:** The spectrum of the frequency illustrates also the effect of attenuation of vibration features of the drilling system partially immersed in a fluid. The results indicate that any interaction of the fluid with the drill string is followed by a moderate *2X . . . 5X* reduction in the critical frequencies and smooth growth of fluctuation where the shape of the orbit patterns is in torus compared to Figure 9b. The results also show increased axial forces in the air as it enters the formation, which can help operators optimize the drilling process and identify potential problems before they become more serious problems.

*5.3. Feature Extraction of the Rubbing Drill String System by Wavelet Synchrosqueezing Transform*

Parametric studies in the context of the dynamics of nonlinear systems require the use of robust and efficient methods for the study of periodic regimes and their stability, such as the use of time–frequency methods. The oscillation feature of the denoising vibration response signal using the wavelet hard-thresholding methodology is first extracted so that the time–frequency resolution is better. Then, the denoised signal is fed to the instantaneous frequency and is taken as an index to qualitatively describe the highly oscillated rotor–casing system and detect the feature of the rub impact in an inviscid fluid through the fluctuation of energy.

The procedure followed to analyze the time signals and interpret the results is different based on the dissipative energy of the system. A direct comparison between conditions with and without friction and fluid is preferred to the main effect plot. Based on the NWSST representation, the influences of some dynamic parameters, including eccentric mass and rub impact under hydrodynamic forces, are investigated as shown in Figure 12. The mass of the fluid affects the frictional impact leading to high quasiperiodic motion. The energy distributions obtained based on the NWSST, as shown in Figure 12a,b, demonstrated that, despite the fluid mass added around the contact point, the friction phenomenon is still perceived by the appearance of some trivial lower amplitude harmonic interference components. The increase in the mass system causes a significant reduction in the bit fluctuation (Figure 12b) where in the severity of stick–slip is more apparent when operating in a dry environment.

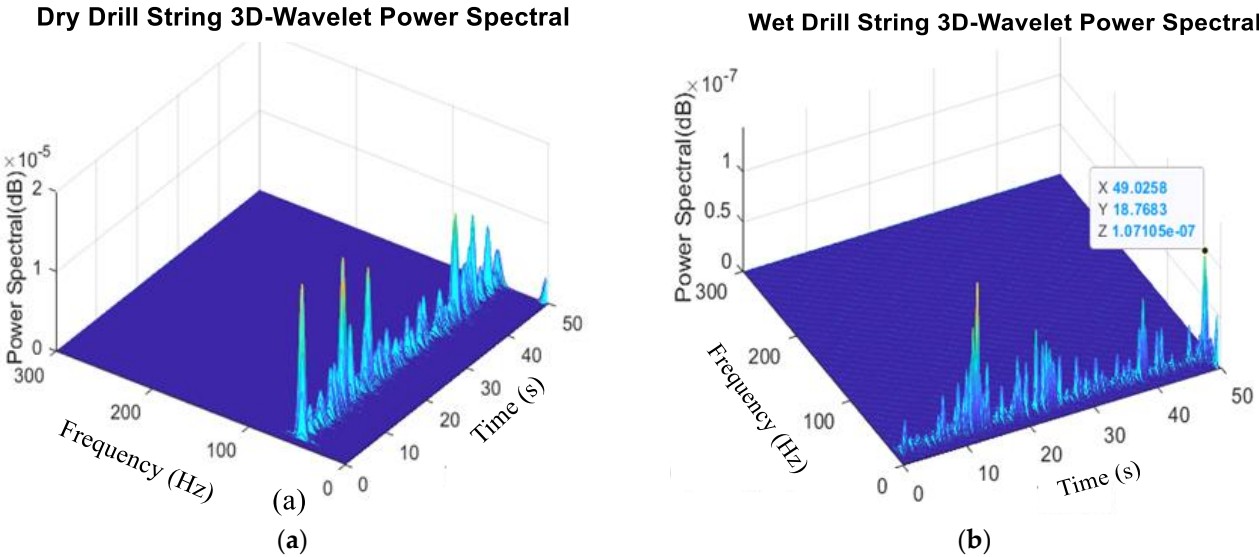

**Figure 12.** No fluid-NWSST representation of the imbalanced drill string signal in the *Y* direction. (**a**) Extracted feature from NWSST; (**b**) Extracted impulse feature from NWSST immersed in fluid.

**Observation 3:** Note that the 3D plot scale is different; the high amplitudes of the 1*X* and 2*X* peaks (t = 0.3 s and 12 s) are lower than the levels observed without fluid impact. The existence of nonsynchronous components in the vibration spectrum is the source of

axial direction and a red flag to the analyst that rubbing problems may exist. The severity of the stick–slip vibrations generates different frequencies based on its physical make-up, as observed in Figure 12, where, despite the presence of the fluid around the stick–slip and drill–wall contacts, the vibrations do not disappear for this drilling section. The drill string system first oscillates with a registering long period of bit sticking with a peak in large resonance amplitudes (Figure 12a), and a sharp decrease in amplitudes of the noiseless extracted IF components reveals the complexity of the time interval between successive high pulses.

Figure 13 illustrates the feature of the rubbing contact when the dissipative inviscid fluid is not integrated into the drill string system. In Figure 13a, the time–frequency ridges from the nonlinear wavelet synchrosqueezing plot can be observed, and this representation is used to obtain a higher resolution for the energy concentration analysis. The maximum energy of the time–frequency ridge is extracted in cycles per sample of the wavelet synchrosqueezed transform. The aperiodic oscillations during the shock reflect the influence of the amplitude of the impact during the contact. During impact, the drill string oscillates at its first resonance frequency of 200 Hz where the frequency of the first resonance mode of the drill string can be observed. The second resonant frequency of the drill string with a high frequency of 400 Hz shows that the drill string, at this time, is excited in its first two modes. The peaks of the second resonance mode of the drill string can be observed with a slight shift below the resonance frequency (circled area) for the second mode, which may be sufficient to change the behavior of the drill string according to its rotational speed. Through time, it can be noted that a satisfactory energy concentration result is observed around the fundamental value of 400 Hz, which denotes a better time–frequency location and better characterization of the time-varying feature. The drill string makes contact with the borehole wall, which then leads to a sustained fluctuating frequency representation. It is worth mentioning, from Figure 13b, that it is difficult to identify the periodicity of a single peak after transforming the signal into the frequency domain. This explains the results of the chaotic behavior obtained in the three-dimensional orbit plot (Figure 10a). Figure 13a shows a distinct set of points that indicates the chaotic behavior of the distributed energy in a fluid medium.

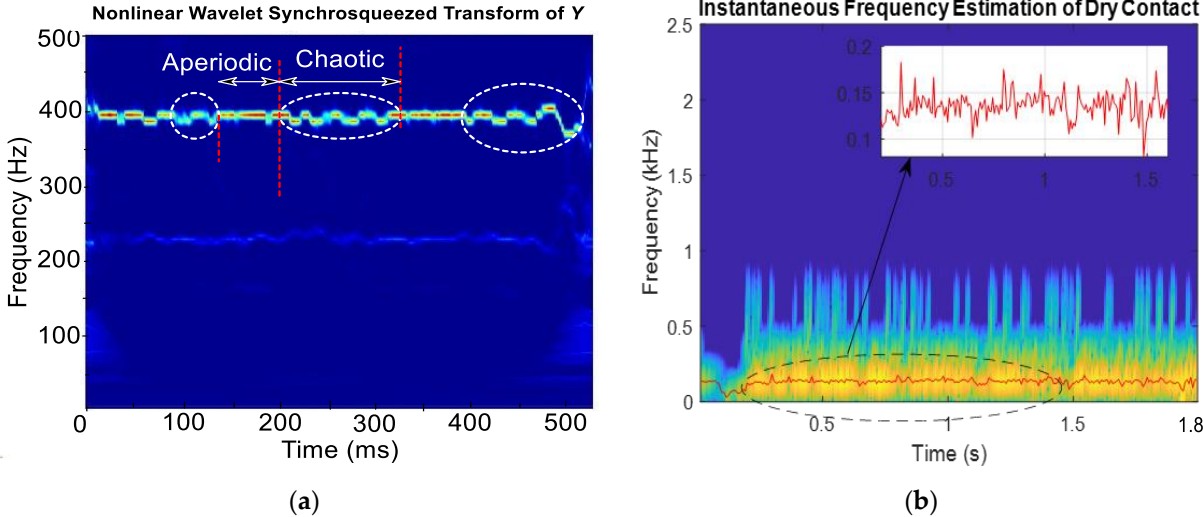

**Figure 13.** No fluid-NWSST representation of the rotor-casing rubbing signal in the *Y* direction. (**a**) Time-frequency ridges from NWSST; (**b**) Estimated IF amplitude spectrum of the signal.

The corresponding estimated frequency spectrum (red line) extracted by the NWSST method shown in Figure 13b contains a regular breaking point that involves the damping effect of the fluid upon shock contact. In Figure 14a, the dominant frequency of the resonance mode of the drill string can be observed at a considerably lower frequency. If the evolution through time of the peak at 250 Hz is followed, it can be seen that there is no longer an oscillation around a defined frequency. However, the curve (circled area) shows

sudden periodic jumps above that of the frequency resonance as if the behavior of the system changed from one mode to another, which corresponds to the minimized chaotic point observed in Figure 13a. In the presence of hydrodynamic forces, the rubbing intensity observed in Figure 13 is attenuated, but the nonlinear phenomena of the fluid introduced a disturbed cascade of peaks throughout the vibration. The IF of the imbalanced system under the frictional impact of the phenomenon is relatively obvious, as observed in Figure 14b, where a series of peak frequency components are aperiodically distributed by irregular low spectral lines representing the contact permanent. As fluid forces and frictional impacts occur, the wavelet scalogram gives a good display of time in the frequency region with an improved analysis of the time-frequency resolution; higher frequency components appear and make it possible to identify the time of occurrence, and the frequency of the friction caused impacts and a sharp decrease in the amplitudes of noiseless extracted IF components, which reveal clearly the time interval between successive high pulses.

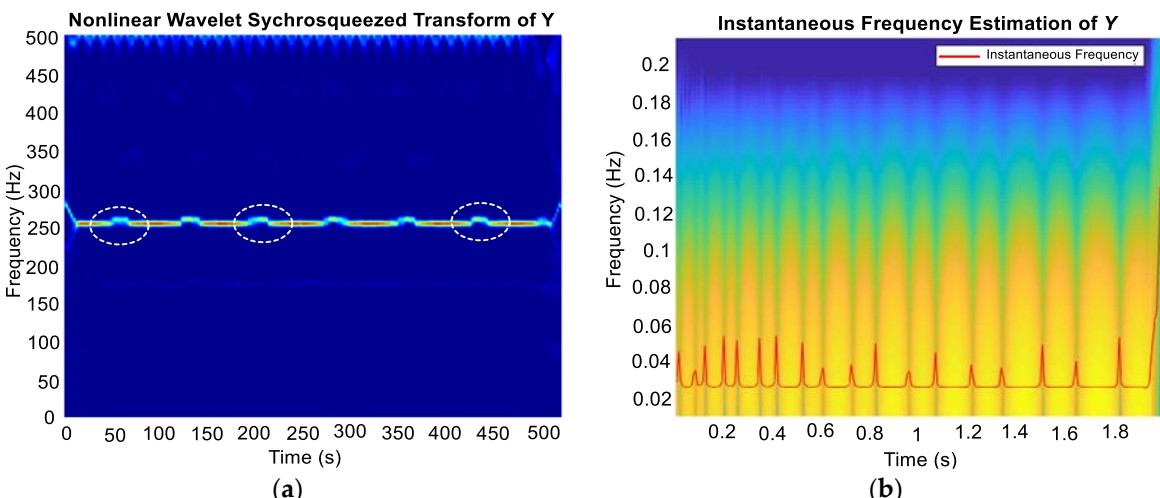

**Figure 14.** Friction/fluid-NWSST representation of the rotor-casing rubbing signal in an inviscid fluid. (**a**) Time-frequency ridges from NWSST; (**b**) Estimated IF amplitude spectrum of the signal.

**Observation 4:** The useful information in the harmonic distribution is observed in Figure 12 at some specific disturbed periods illustrated by the high peak's shape disturbance throughout the signal, while in the spectrum bands, the diminution of the impact amplitude is prominent. It can also be seen from Figure 12 that there are good approximations of the signal's instantaneous frequency, which oscillates periodically and differs from the IF of the imbalanced drilling/casing system with the eliminated hydrodynamic forces. The damping imposed by the presence of fluid makes it possible to obtain a good prediction of the behavior of the drill string subjected to an impact. Adding a dissipative inviscid fluid in the system minimizes its vibration by significantly amortizing the shock. It is obvious that, if only the imbalance effect is taken into account, the IF remains unchanged, while the shock generated by the rotor–casing system oscillates gradually in amplitude.

## 6. Experimental Analysis and Discussion

In this section, an extracted signal contaminated with strong noise is used to evaluate the performance of the NWSST technique. The drill string vibration signal under various operating conditions is imported as an excel sheet into Matlab to perform the time–frequency analysis. Although it may be useful to experimentally study the effects of the fluid on the dynamic behavior of the rotor–casing, the reconstructed signal with multiple faults is first denoised using a wavelet thresholding technique through the original signal. A detailed practical study of NWSST for rub detection and monitoring in rotors is illustrated using two sets of experimental data (with and without fluid forces) acquired under varying speed conditions.

### 6.1. Practical Test 1: Imbalanced Vertical Shaft Response

From the FFT characteristics (Figure 15b), it is evident that the resonant frequency is higher than in the case of the imbalanced drill string operating in water in Figure 15a. The experimental results demonstrate that, during start-up, the lateral trajectory of the drill string increases gradually, and its circular orbit consists of mixing two harmonic paths of different frequencies (Figures 15a and 16a). The orbital path rotates at a considerably lower amplitude (note the different magnitudes in the X- and Y-axes in the figures after immersion) as shown in Figures 15b and 16b.

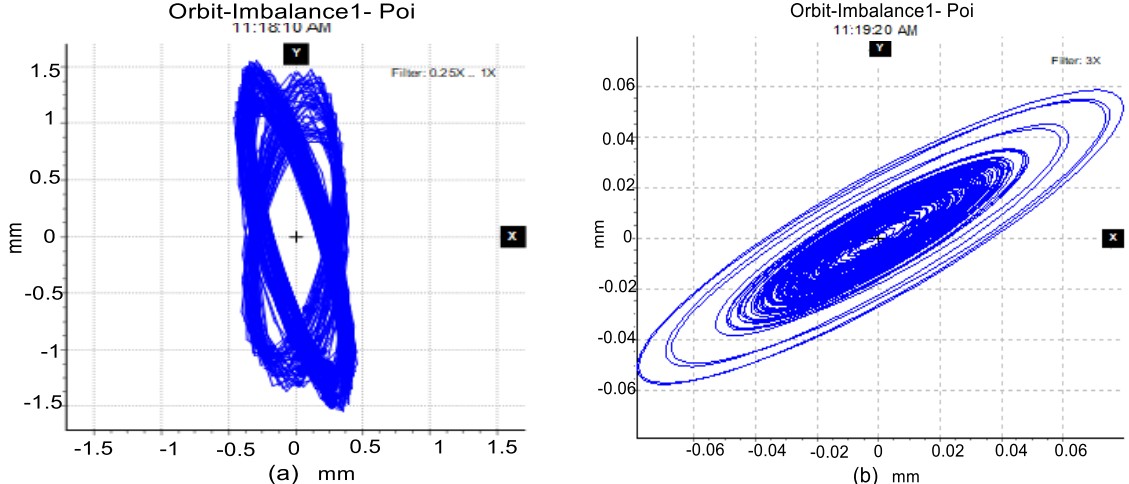

**Figure 15.** Experimental orbit response of the imbalanced drill string in fluid (**a**) without water and (**b**) with water.

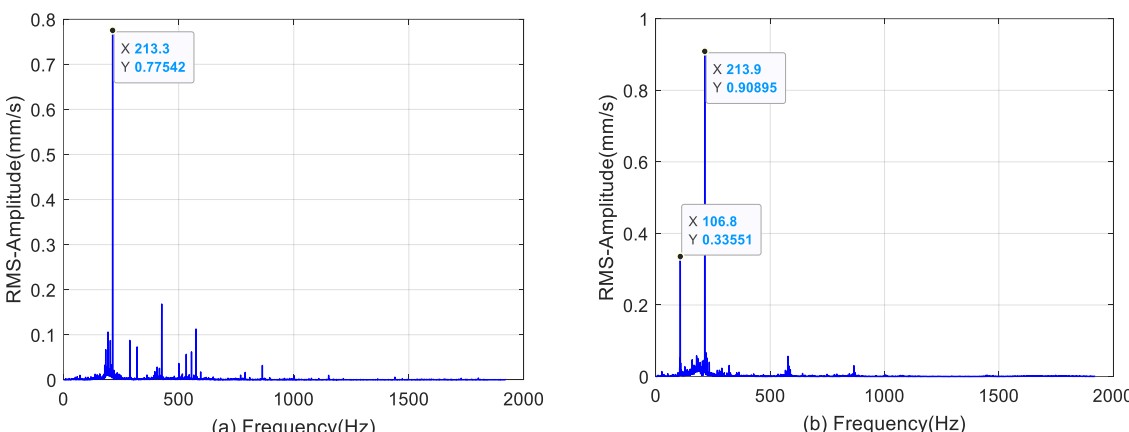

**Figure 16.** Experimental frequency response of the imbalanced rotor (**a**) without water and (**b**) with water.

For the experiment presented, the drill string rotates in the air, and for the fluid, at a speed of approximately 1600 rpm with an imbalanced mass, the progressive excitation leads to rubbing contact. Due to the constraint on the axial displacement, measurement probes (Figure 6d) to detect the behavior of the drill string downhole were used for the actual drill string and to extract the characteristics when some unwanted vibration occurred in the fluid medium. The lateral vibrations of an imbalanced swirling drill string at critical velocities and influenced by fluid forces were measured, and the impulse characteristics were then extracted and plotted as shown in Figures 17 and 18.

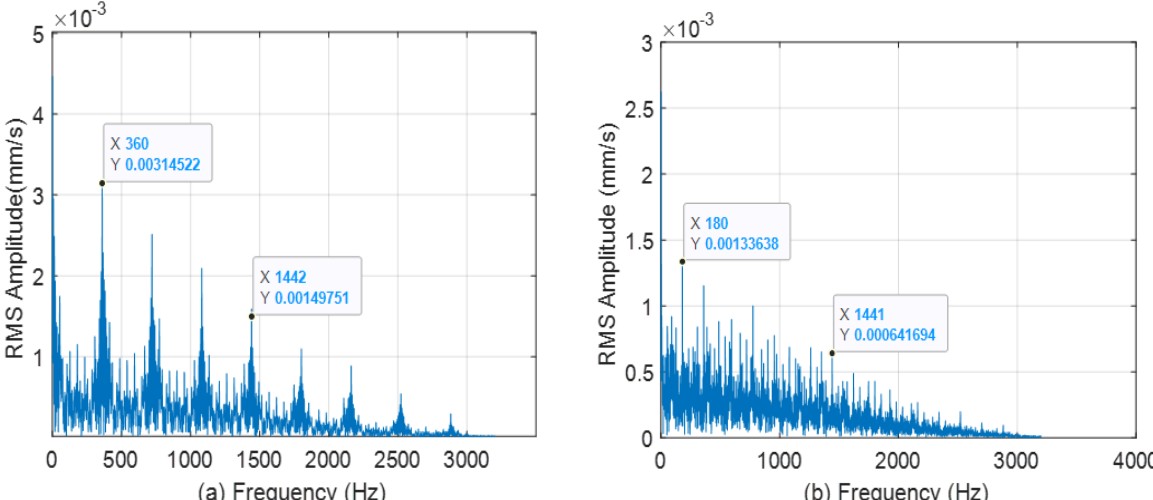

**Figure 17.** Experimental response of imbalance and rotor-casing rub (**a**) without water and (**b**) with water.

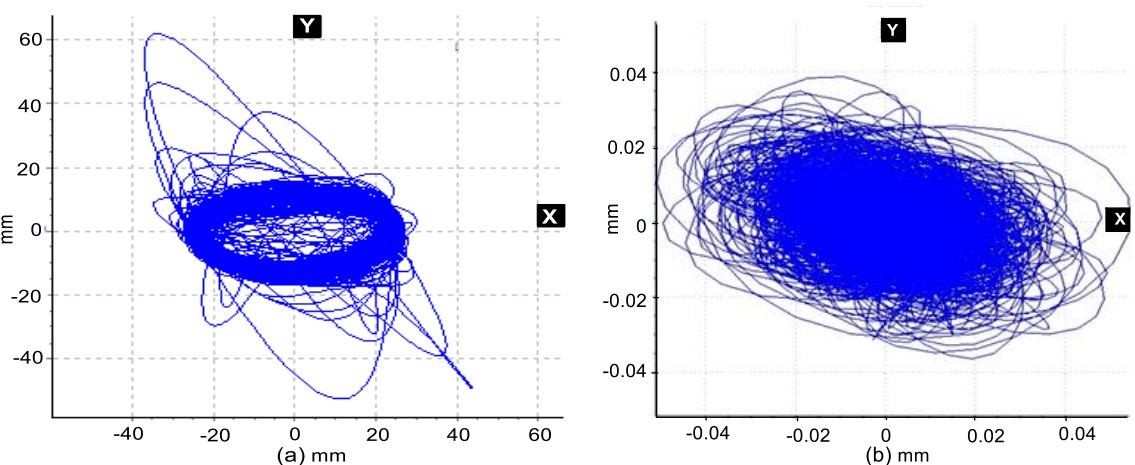

**Figure 18.** Experimental orbit response of imbalanced rotor-casing rub (**a**) without water and (**b**) with water.

### 6.2. Practical Test 2: Imbalanced Vertical Shaft with Friction and Fluid

Figures 17a and 18a are almost similar to the behavior of that in the fluid. The frictional impact of higher resonance frequencies and larger subharmonic peaks occurring with the bounce of the outer drill string orbit can be seen in both Figures 17 and 18. In general, friction-induced noise and vibrations exhibit nonlinear, time-varying transient characteristics. For comparison, it is shown that the obtained experimental signal is highly consistent with the simulated defect signal. The extracted impulse features in the inviscid fluid are plotted as shown in Figure 17b. The proposed results can accurately display the impulse characteristics of the rubbing contact, even when the fault signal is buried under strong noise interference.

**Observation 5:** For the case where faults such as imbalance and rotor–stator rub coexist, the response of the drill string orbit in the lateral direction and its FFT spectrum showed that the harmonics of $1 \times$ order (213.8 Hz) representing the imbalance exist in each response (with and without fluid). The FFT spectral plot also shows that higher frequencies, for example, 2X, 3X, 5X, etc., get excited as it occurs. The presence of a weak subharmonic resonance at 1/2 of the critical bending speed can be observed in the submerged imbalanced rubbing drill string. The experimental results show how the fluid also indirectly affects the frictional torque in the system.

    In this section, since the rub fault is highly nonlinear and the fluid expression is always transient, the time–frequency representation (TFR) is properly used for this study using both IF and NWSST. Figures 19 and 20 show waveform plots of the IFs extracted by wavelet synchrosqueezing and the NWSST spectra without and with fluid, respectively. In general, aperiodic and chaotic windows can be observed in the experimental responses in Figure 19a, and a period-doubling route can be noted from the expanded diagram, as presented in Figure 20a and observed previously in Figures 13a and 14a. To further experimentally show the performance of the NWSST in analyzing the signal with multiple faults, the center frequencies of these fault components are confined to the interval [90 Hz to 500 Hz] where the TF characteristics of these two faults are captured by the NWSST (Figure 19a). The NWSST provides a time–frequency representation of the signal, which may require further analysis and interpretation to extract meaningful information about the dynamic behavior of the system. The corresponding IF is then used to filter these two fault signals. It can be seen in Figure 19b that the reconstructed IF with a denoised signal is highly consistent. With the increase in the impact, the large number of continuous components of the frequency spectrum of the system shows that there is a serious frictional impact phenomenon in the system. These frequency components can be used as characteristic frequencies for drill string rub impact fault diagnosis. Figure 21 shows the standard deviation trends corresponding to the defective cases of the selected signal segments, revealing an explicit energy level difference between the reference (air) and fluid medium (water).

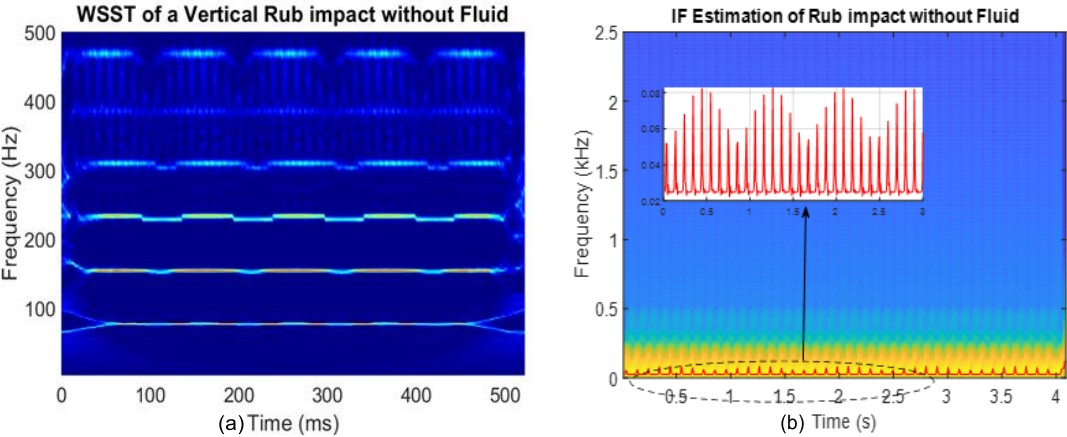

**Figure 19.** Time–frequency map of imbalanced rotor-casing rub; (**a**) NWSST response; (**b**) IF response.

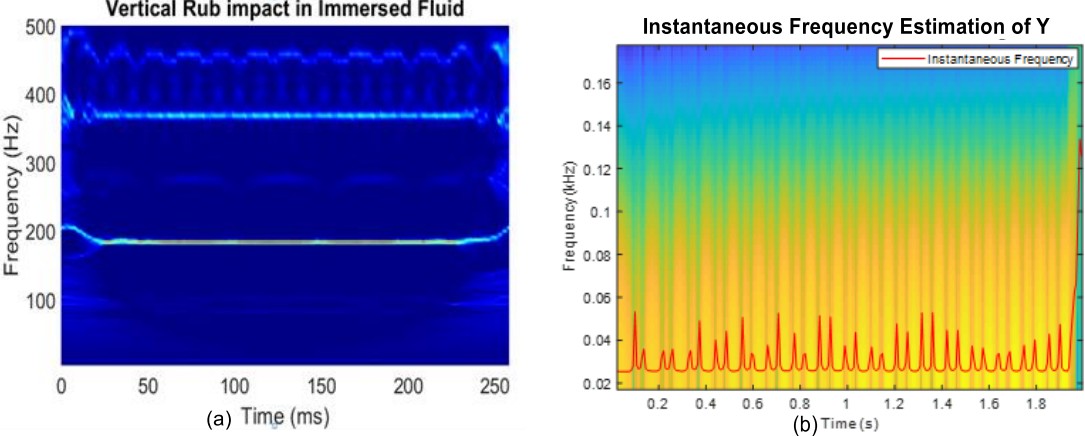

**Figure 20.** Time-frequency map of imbalanced drill string-casing rub in fluid; (**a**) NWSST response; (**b**) IF response.

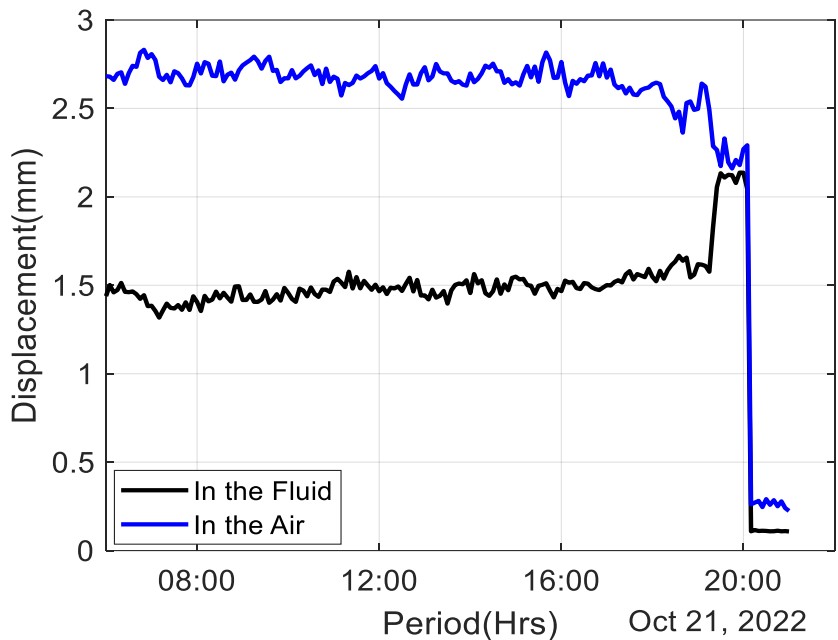

**Figure 21.** The overall trend of the drill string vibration level. The blue line shows the transient contact in the air with a coefficient of friction μ = 0.2, and the black line shows the frictional contact inside the fluid with the same coefficient of friction.

**Observation 6:** Figure 20b shows that the main difference in the dynamic response with and without the fluid–structure interaction model lies in the dynamic amplitude of the response and the major influence of the fluid in the side frequencies on the side mode shape. The overall trend is different for the two models when the vibration amplitude curve of the drill string without fluid is greater than that in water. To track the operating conditions of the drill string and the trend deterioration to predict the downtime, it can be seen from the overall measurement that the fluid-free vibration amplitude suddenly increases (Figure 21). The level of vibration on impact is still normal, so no action needs to be initiated, but the vibration should be measured regularly from then on, and any increase in amplitude should be accurately assessed.

## 7. Conclusions

In this research, the common frictional defect between a drill string and a stator was modeled and studied considering the whole torque, centrifugal force, stabilizer, bit–rock interactions, shock and friction between the stabilizer and the borehole wall, and fluid–structure interactions (which circulate around the drill shaft). The conventional FFT spectrum and orbit patterns provide preliminary results for machine operators to identify the exact occurrence time and frequency of these rubbing-caused impacts in the fluid. The instantaneous frequency based on the nonlinear wavelet synchronized transform creates significant interference sufficient to obtain a good resolution of the impacts caused by friction. The conditions of shock are difficult to separate; in particular, for the important frictions, the tests of the simulated and experimental signals show that the period of the shock is very difficult to determine. The fluid has a major influence on the lateral frequency, but the axial frequency and torsional frequency are not affected. Moreover, the results also show that the synchrosqueezed techniques significantly improved the readability of the results by increasing the time of occurrence and the period of high-frequency impact during friction in a fluid medium. The existence and severity of friction inside the inviscid liquid becomes detectable. Synchrosqueezed methods are effective for estimating and detecting the instantaneous frequencies of lower amplitude frictional impacts as well as the behavior of the drill string systems in an inviscid fluid. However, it only extracts nonlinear features, which may not be sufficient for some applications requiring more complete information. Their limitations need to be carefully considered and addressed to ensure accurate

and reliable results. However, the accurate measurement of the axial contact of the drill string and the rock poses a great challenge for experimental fault diagnosis, which is a limitation of the present research. This forms the motivation for future research work in addition to experimentally studying the effects of frictional impacts in viscous and inviscid fluids in both healthy and defective drill strings and then correlating the results with those of the proposed theoretical model.

**Author Contributions:** The topic of this research was conceptualized by B.X.T.K., D.F.S. and A.A.A.; formal analysis was performed by B.X.T.K.; the first version of the manuscript was prepared by B.X.T.K.; this version of the manuscript was read and approved by B.X.T.K., D.F.S. and A.A.A. who also substantially contributed to it. All authors have read and agreed to the published version of the manuscript.

**Funding:** This research received no external funding.

**Institutional Review Board Statement:** Not applicable.

**Informed Consent Statement:** Not applicable.

**Data Availability Statement:** Not applicable.

**Acknowledgments:** This work is based on the research supported in part by the Vaal University of Technology (VUT), South Africa, through the Department of Higher Education and Training University Capacity Development Grant.

**Conflicts of Interest:** The authors declare no conflict of interest.

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
