# Peer review of "Modeling and Analysis of Drill String–Casing Collision under the Influence of Inviscid Fluid Forces"

_applsci, doi:10.3390/app13063557_

Round 1
Reviewer 1 Report
Please redo the figures 6, 7, 8 and 18.
Author Response
The authors would like to thank the manuscript reviewers for their constructive and professional comments and suggestions, which were very helpful in improving the paper. Additional explanations and analyses were introduced in the amended version of the article.

Reviewer 2 Report
This manuscript, “Modeling and Analysis of Drill string-Casing Collision Under the Influence of Inviscid Fluid Forces”, aimed to analyze the drill string casing system under the influence of inviscid fluid forces. Comments are given below.
1. It is recommended that more references be added.
2. Figure 1 (a) text colour needs to be changed.
3. The manuscript is inconsistent in some of its typefaces, such as the letter Y in line 228 and line 231, etc.
4. Font and symbol formats in figures and tables need to be standardised.
5. Line 633, where is the answer of “But what will the axial data reveal?”.
6. Figure 13 is missing the name of the X coordinate and the colour of the text in the figure is recommended to be changed.
7. More care should be taken to avoid such issues. For example, Line 227 is missing the right bracket; Line 761, “Figures 17” deletes ‘s’; Line 769, Table 17 is changed to Figure 17; Line 810, Observation 5 changes to 6.
Author Response

(The authors gave the same response as above.)

Reviewer 3 Report
It can be said that the theoretical introduction is exhaustive, but it would be good to expand the literature review to include recent items of international scope. The second paragraph is very long and addresses diverse issues. I suggest splitting it into several paragraphs. Please consider shortening the article - perhaps splitting it into 2 or even 3? The article has the character of exhaustive research topics - it is interesting from an engineering point of view, but by its volume it is quite heavy to read.
Mathematical modeling issues described correctly and comprehensively. On top of that, understandably. All the questions that arise are explained in the text. Drawing self-explanatory. I think that the theta symbol is not explained - I am looking for an explanation, but unfortunately I do not see it.
Line 383 - please watch for symbols in italics.
Please explain why such a value of friction coefficient - its value is puzzling. Drawings very clear and understandable. Does the NWSST method have any limitations of applicability?
Author Response

(The authors gave the same response as above.)
